# Impact of Olive Oil Components on the Expression of Genes Related to Type 2 Diabetes Mellitus

**DOI:** 10.3390/nu17030570

**Published:** 2025-02-03

**Authors:** Camelia Munteanu, Polina Kotova, Betty Schwartz

**Affiliations:** 1Department of Plant Culture, Faculty of Agriculture, University of Agricultural Sciences and Veterinary Medicine of Cluj-Napoca, 400372 Cluj-Napoca, Romania; 2The Institute of Biochemistry, Food Science and Nutrition, The School of Nutritional Sciences, Robert H. Smith Faculty of Agriculture, Food and Environment, The Hebrew University of Jerusalem, Rehovot 9190500, Israel

**Keywords:** olive oil, type 2 diabetes mellitus, oxidative stress, inflammation, polyphenols

## Abstract

Type 2 diabetes mellitus (T2DM) is a multifactorial metabolic disorder characterized by insulin resistance and beta cell dysfunction, resulting in hyperglycemia. Olive oil, a cornerstone of the Mediterranean diet, has attracted considerable attention due to its potential health benefits, including reducing the risk of developing T2DM. This literature review aims to critically examine and synthesize existing research regarding the impact of olive oil on the expression of genes relevant to T2DM. This paper also seeks to provide an immunological and genetic perspective on the signaling pathways of the main components of extra virgin olive oil. Key bioactive components of olive oil, such as oleic acid and phenolic compounds, were identified as modulators of insulin signaling. These compounds enhanced the insulin signaling pathway, improved lipid metabolism, and reduced oxidative stress by decreasing reactive oxygen species (ROS) production. Additionally, they were shown to alleviate inflammation by inhibiting the NF-κB pathway and downregulating pro-inflammatory cytokines and enzymes. Furthermore, these bioactive compounds were observed to mitigate endoplasmic reticulum (ER) stress by downregulating stress markers, thereby protecting beta cells from apoptosis and preserving their function. In summary, olive oil, particularly its bioactive constituents, has been demonstrated to enhance insulin sensitivity, protect beta cell function, and reduce inflammation and oxidative stress by modulating key genes involved in these processes. These findings underscore olive oil’s therapeutic potential in managing T2DM. However, further research, including well-designed human clinical trials, is required to fully elucidate the role of olive oil in personalized nutrition strategies for the prevention and treatment of T2DM.

## 1. Introduction

The global prevalence of type 2 diabetes mellitus (T2DM) is rising at an alarming rate [1]. Lifestyle factors, particularly dietary habits, are critical in both the prevention and management of the disease, with the primary goal being effective glycemic control or the delay of chronic diabetic complications and management of the disease aimed at achieving an adequate glycemic control or at delaying the onset of diabetic chronic complications [2]. Mediterranean diet (MD), which prominently features olive oil as the primary source of fat, has long been associated with favorable outcomes in the prevention and management of T2D [3]. In this context, olive oil consumption is considered a key determinant of the health benefits attributed to the MD [4]. Olive oil is rich in monounsaturated fatty acids (MUFAs) and contains various micronutrients, including phenolic compounds. Research has shown that the phenolic fraction of olive oil exerts beneficial effects on oxidative stress and inflammation, partly through the modulation of gene expression involved in these processes [5]. Recent studies have focused on the health-promoting properties of olive oil, with particular attention to the concentration of polyphenols, which appears to be crucial in establishing a cause-and-effect relationship [6]. Despite growing evidence supporting the beneficial effects of the MD and its components on T2DM, the precise mechanisms underlying these effects remain only partially understood. In the present review, we will address the effect of all components of olive oil on the amelioration of T2DM.

Also, it is important to mention that pathogens, weeds, and insects can reduce crop yields by 25% to 50%. Synthetic pesticides shield plants or plant products from dangerous organisms or prevent them from acting. Pest management increases the amount of food produced worldwide [7]. “Plant protection products” (PPP) refer to commercial formulations of pesticides that contain active ingredients, safeners, or synergists. To prevent diseases or pest assaults that weaken the trees and render the olives unfit for human consumption or of lower quality, olive groves require ongoing, efficient maintenance and treatment. Furthermore, olive harvests have been impacted by the olive fly and current climate conditions [8]. Due to these factors, PPP is frequently used in olive groves to guarantee crop output and harvest yield. To ensure that olive trees produce high-quality fruit with an ideal quantitative yield, it is crucial to understand when and how PPP should be applied. Common fungicides for olive trees include tebuconazol and copper oxychloride. Insecticides (such as lambda-cyhalothrin, deltamethrin, dimethoate, phosmet, and spinosad) and herbicides (such as glyphosate, flazasulfuron, and other sulfonylurea chemicals) are other PPPs that are frequently employed [9]. If these chemicals are used excessively in olive oil, it can harm human health, particularly if the specified dosages and application schedules are not followed. Following Regulation (EC) No 400/2014, reporting EU countries examined 10,884 samples of 11 distinct food products as part of the 2015 European Union Coordinated Program (EUCP). Specifically, out of the 1045 olive oil samples examined, 84.5% had no detectable pesticide residues, whereas 15.5% had one or more pesticides in detectable amounts. Notably, 4.2% of samples of olive oil had several residues. Compared to 2012, when pesticide residues were found in 22% of olive oil samples, the overall quantification rate was somewhat lower. The olive oil samples that were examined contained 29 distinct pesticides at amounts that were either equal to or higher than the limit of quantification (LOQ) [9].

### 1.1. Type 2 Diabetes Mellitus (T2DM)

T2DM represents a significant public health challenge and is one of the most serious chronic metabolic disorders worldwide. It not only adversely affects individual health but also imposes a considerable economic burden on healthcare systems [10]. According to the International Diabetes Federation, the estimated prevalence of diabetes among adults aged 20 to 79 years was approximately 537 million individuals globally in 2021, representing 10.5% of the global population. This statistic indicates that more than one in ten adults is affected by this metabolic disease. Projections suggest that the number of individuals with diabetes will rise to 643 million by 2030 and 783 million by 2045. In 2021 alone, diabetes was responsible for 6.7 million deaths and incurred healthcare costs of at least 966 billion dollars, reflecting a staggering 316% increase in health expenditures over the past 15 years. T2DM constitutes the most prevalent form of diabetes, accounting for over 90% of all diabetes cases worldwide [11].

The mechanisms underlying the development of T2DM are the subject of intensive research, recognized as stemming from a complex interplay of genetic predisposition alongside lifestyle and environmental factors, such as overweight and physical inactivity [12,13]. Gaining insight into these underlying mechanisms is essential for the formulation of effective preventive and therapeutic strategies. Recent studies indicate that a diverse range of dietary factors, particularly those abundant in bioactive compounds, may influence the expression of genes associated with T2DM [13,14].

### 1.2. Pathogenesis of Type 2 Diabetes Mellitus (T2DM)

T2DM is a multifaceted metabolic disorder characterized by chronic hyperglycemia, resulting from a combination of insulin resistance and beta cell dysfunction. The pathophysiology of T2DM encompasses several interrelated mechanisms that disrupt glucose homeostasis, ultimately culminating in the clinical manifestations of the disease [15].

A fundamental aspect of T2DM is insulin resistance, which primarily affects skeletal muscle, liver, and adipose tissue. Insulin resistance is characterized by a diminished cellular response to insulin [16]. In healthy individuals, insulin facilitates glucose uptake into cells, particularly in muscle and adipose tissues, while simultaneously suppressing hepatic glucose production. However, during insulin resistance, these regulatory processes are impaired, leading to elevated blood glucose levels [17].

Beta cell dysfunction represents another critical element in the pathogenesis of T2DM. In individuals predisposed to the disease, both genetic and environmental factors compromise beta cell function, resulting in reduced insulin secretion [16]. Initially, beta cells attempt to compensate for insulin resistance by increasing insulin production; however, this compensatory mechanism eventually fails over time, leading to diminished insulin secretion [17].

The interplay between insulin resistance and beta cell dysfunction is central to the development of T2DM. Insulin resistance creates an increased demand for insulin secretion from beta cells. When these cells are unable to meet the heightened demand due to intrinsic defects or acquired impairments, this results in decreased insulin production and overt hyperglycemia—hallmark features of T2DM [18], see Figure 1.

The pathogenesis of type 2 diabetes mellitus (T2DM) involves genetic, environmental, and metabolic factors that disrupt normal glucose regulation. One of the main characteristics of muscle, fat, and liver cells is insulin resistance, which occurs when the cells lose their sensitivity to insulin. The liver’s increased hepatic glucose synthesis worsens elevated blood glucose levels. Because inflammatory cytokines from visceral fat disrupt insulin signaling, inflammation plays a role in insulin resistance. An imbalance of reactive oxygen species (ROS) causes oxidative stress, further impairing insulin function and encouraging cellular damage. Insulin resistance and increased fat formation are two consequences of mitochondrial dysfunction, which affects how cells use energy. By releasing fatty acids and inflammatory chemicals, obesity, especially abdominal fat, exacerbates insulin resistance. Environmental factors that cause and exacerbate metabolic abnormalities include a flawed diet high in carbohydrates and fats and a lack of physical activity. Genetic predisposition can affect how the body reacts to external stimuli, making people more likely to develop insulin resistance. T2DM is further exacerbated by the disruption of the gut flora, which has been connected to inflammation and poor glucose metabolism. Pancreatic β-cell dysfunction develops over time, affecting insulin production and resulting in chronic hyperglycemia, a defining feature of type 2 diabetes.

### 1.3. Mechanisms Leading to Insulin Resistance

At the molecular level, insulin resistance is characterized by defects in the insulin signaling pathway, which begins with insulin binding to its receptor on the cell surface. The insulin receptor consists of two extracellular α-subunits and two intracellular β-subunits. Upon binding, the receptor undergoes autophosphorylation, activating the intrinsic tyrosine kinase activity of the receptor by phosphorylating specific tyrosine residues on the β-subunits. This activation enables the receptor to phosphorylate insulin receptor substrate (IRS) proteins on their tyrosine residues, particularly IRS-1 and IRS-2. These phosphorylated IRS proteins then create docking sites for downstream signaling molecules, including phosphoinositide 3-kinase (PI3K), crucial in mediating insulin’s effects on glucose metabolism [19].

Activation of PI3K catalyzes the conversion of phosphatidylinositol (4,5)-bisphosphate (PIP2) to phosphatidylinositol (3,4,5)-trisphosphate (PIP3), a crucial secondary messenger. PIP3 facilitates the recruitment and activation of protein kinase B (Akt) to the plasma membrane. Once activated, Akt plays a vital role in metabolic regulation by promoting glucose uptake through the translocation of glucose transporter Type 4 (GLUT4) to the cell membrane [19,20].

In the context of insulin resistance, various defects compromise the normal functioning of the insulin signaling pathway, leading to impaired glucose uptake and metabolism. A significant alteration observed in this condition is the reduced tyrosine phosphorylation of IRS proteins, which is essential for effective downstream signaling. Simultaneously, there is an increase in serine/threonine phosphorylation of IRS proteins, further disrupting the insulin signaling cascade. These modifications hinder the activation of PI3K and Akt, critically impairing the insulin signaling pathway and inhibiting GLUT4 translocation [19]. Factors contributing to the increased serine/threonine phosphorylation of IRS proteins include inflammatory cytokines, lipid intermediates, oxidative stress, and endoplasmic reticulum (ER) stress [21].

Excessive intake of dietary carbohydrates and fats, particularly saturated fats, is a well-documented contributor to the development of insulin resistance. This excess may disrupt insulin signaling pathways through multiple interconnected mechanisms, including lipid accumulation, oxidative stress, ER stress, and chronic inflammation [22,23].

Dysregulation of lipid metabolism plays a critical role in the onset of insulin resistance [22]. High-fat diets can lead to excessive accumulation of fatty acids in non-adipose tissues, such as the liver and skeletal muscle. This phenomenon, known as lipotoxicity, significantly contributes to the development of insulin resistance [22,23]. Accumulation of lipid intermediates, such as diacylglycerols (DAGs) and ceramides, in these tissues interferes with insulin signaling pathways. For example, DAGs can activate protein kinase C (PKC) isoforms, which phosphorylate IRS proteins on serine and threonine residues [24].

Another source of these lipid intermediates is the impaired oxidation of fatty acids within mitochondria. When there is an excess of fatty acids, the mitochondrial capacity to oxidize these fats may become overwhelmed, resulting in the accumulation of lipid intermediates and increased production of reactive oxygen species (ROS) [23]. The resultant ROS can induce oxidative stress, damaging cellular components involved in insulin signaling [22].

Chronic low-grade inflammation also significantly contributes to insulin resistance by increasing the serine/threonine phosphorylation of IRS proteins [25]. In obesity-related insulin resistance, hypertrophic adipocytes secrete elevated levels of pro-inflammatory cytokines, such as tumor necrosis factor-alpha (TNF-α) and interleukin-6 (IL-6). These cytokines activate serine kinases, including c-Jun *N*-terminal kinase (JNK) and IκB kinase (IKK), which phosphorylate IRS proteins on serine residues rather than on tyrosine residues [20,22].

It has been demonstrated that consuming too much high fructose, especially from processed foods and sugary drinks, can lead to insulin resistance, which is a risk factor for type 2 diabetes [26]. Fructose is mainly digested in the liver, which can be turned into fat and cause non-alcoholic fatty liver disease (NAFLD), in contrast to glucose, which is metabolized in other tissues. This fat buildup exacerbates insulin resistance by compromising the liver’s capacity to control blood glucose [27]. Type 2 diabetes is more likely to develop because the body’s cells are less sensitive to insulin over time, necessitating higher hormone levels to regulate blood glucose. Fruits naturally contain fructose, although entire fruits contain far less of it. They can lessen their adverse effects on metabolism because fiber, vitamins, and antioxidants accompany them [28].

Conversely, more metabolic hazards are associated with the concentrated form of fructose found in added sugars, especially high-fructose corn syrup (HFCS). According to studies, diets high in HFCS might exacerbate oxidative stress and inflammation, leading to insulin resistance [29]. Thus, lowering the consumption of processed foods and sweetened beverages with high added fructose content is essential for controlling and avoiding insulin resistance and type 2 diabetes, even though eating fruits is not a significant worry for insulin resistance [30].

### 1.4. Mechanisms Leading to Beta Cell Dysfunction

Beta cell dysfunction is the result of a confluence of genetic predispositions, environmental influences, and metabolic stressors [16,31]. Various molecular pathways implicated in this dysfunction include those associated with the unfolded protein response (UPR) to endoplasmic reticulum (ER) stress, mitochondrial dysfunction, and inflammatory signaling pathways [32].

Key metabolic stressors contributing to beta cell dysfunction include chronic hyperglycemia (glucotoxicity) and elevated free fatty acids (lipotoxicity) [33]. Glucotoxicity arises from prolonged exposure to elevated glucose levels, which impair insulin gene expression and overall beta cell function. Conversely, lipotoxicity results from excessive fatty acid accumulation, leading to beta cell apoptosis through mechanisms involving oxidative stress, mitochondrial dysfunction, and ER stress [34].

ER stress plays a pivotal role in the molecular mechanisms leading to beta cell dysfunction [35]. The ER is crucial for the proper folding and processing of proteins, including insulin [36]. Under conditions of metabolic stress, the increased demand for insulin synthesis can overwhelm the protein-folding capacity of the ER, resulting in the accumulation of misfolded or unfolded proteins. This situation triggers the UPR, a cellular stress response aimed at restoring ER homeostasis [32,37].

The UPR is a vital mechanism in beta cells, designed to maintain ER function and ensure proper protein folding [38]. It comprises three primary signaling pathways mediated by ER membrane proteins: inositol-requiring enzyme 1 (IRE1), protein kinase R-like ER kinase (PERK), and activating transcription factor 6 (ATF6). Each pathway contributes uniquely to the adaptive and, under chronic stress conditions, apoptotic responses of beta cells [39]. In the absence of stress, IRE1, ATF6, and PERK exist as inactive monomers bound to the ER chaperone immunoglobulin heavy chain-binding protein (BiP). During ER stress, BiP dissociates from these proteins, triggering the initiation of the UPR [21].

Activation of PERK leads to the phosphorylation of eukaryotic initiation factor 2 alpha (eIF2α), which attenuates protein translation, thereby decreasing the influx of nascent proteins into the ER. This reduction alleviates the burden of misfolded proteins. Concurrently, phosphorylated eIF2α selectively enhances the translation of activating transcription factor 4 (ATF4), a transcription factor that upregulates genes involved in amino acid metabolism, oxidative stress response, and apoptosis. In beta cells, PERK signaling assists in managing acute ER stress, but under persistent stress, it can promote the expression of the CCAAT-enhancer-binding protein (C/EBP) homologous protein (CHOP), a pro-apoptotic factor that drives beta cell death [21,34].

Upon activation, IRE1 undergoes autophosphorylation and oligomerization, which triggers its endoribonuclease activity. This activity cleaves X-box binding protein 1 (XBP1) mRNA, producing the spliced form XBP1s, a transcription factor. XBP1s then translocate to the nucleus, which induces the expression of genes involved in protein folding and ER-associated protein degradation (ERAD), key processes in maintaining cellular homeostasis and alleviating ER stress. In beta cells, IRE1-XBP1 signaling enhances the capacity to manage the high load of proinsulin processing [34,36]. However, chronic activation of IRE1 can also activate apoptotic pathways via the JNK and MAPK (mitogen-activated protein kinase) pathways, contributing to beta cell apoptosis [37].

In response to ER stress, ATF6 translocates from the ER to the Golgi apparatus, where it is cleaved by site-1 and site-2 proteases. The cleaved cytosolic fragment of ATF6 (ATF6f) functions as a transcription factor that upregulates genes involved in protein folding and ERAD. By enhancing the expression of chaperones and components of the ERAD pathway, ATF6 improves protein folding and the degradation of misfolded proteins. Nonetheless, like the other UPR pathways, chronic activation of ATF6 can also contribute to beta cell dysfunction and apoptosis through the activation of pro-apoptotic signaling mediators such as CHOP and the JNK and p38 MAPK pathways [32,34].

The unfolded protein response (UPR) in beta cells acts as a double-edged sword. While it is crucial for adapting to the physiological demands of insulin production and managing acute endoplasmic reticulum (ER) stress, chronic or excessive activation of the UPR can lead to beta cell apoptosis. This paradox arises because prolonged ER stress can exceed the adaptive capacity of the UPR, causing a shift toward cell death pathways. This shift becomes particularly significant in the context of type 2 diabetes, where chronic hyperglycemia and elevated free fatty acids continuously induce ER stress in beta cells, ultimately contributing to their dysfunction and loss [38].

Oxidative stress is another critical contributor to beta cell dysfunction. Excessive production of reactive oxygen species (ROS) during chronic hyperglycemia and lipotoxicity can damage cellular components, including DNA, proteins, and membrane lipids, thereby promoting mitochondrial dysfunction and accelerating apoptosis while reducing beta cell mass through the activation of pro-inflammatory mechanisms involving JNK and MAPK pathways [40,41].

Inflammatory signaling also plays a significant role in beta cell dysfunction. Pro-inflammatory cytokines such as interleukin-1 beta (IL-1β), tumor necrosis factor-alpha (TNF-α), and interferon-gamma (IFN-γ) activate intracellular signaling cascades, including the JNK and MAPK pathways, leading to the expression of genes involved in inflammation and apoptosis [40].

The mechanisms underlying beta cell dysfunction in T2DM are intricately interconnected, with ER stress, oxidative stress, inflammation, and mitochondrial dysfunction forming a vicious cycle that exacerbates beta cell impairment. ER stress triggers the UPR, which, if unresolved, can lead to beta cell apoptosis. This stress is compounded by oxidative stress, where an overproduction of ROS damages cellular components and further disrupts protein folding in the ER. Inflammatory cytokines, such as TNF-α and IL-1β, amplify both ER stress and oxidative stress by activating signaling pathways that induce apoptosis and inflammatory responses. Concurrently, mitochondrial dysfunction impairs ATP production and increases ROS generation, further exacerbating both ER and oxidative stress. Together, these stressors create a feedback loop that perpetuates beta cell dysfunction, ultimately contributing to the progression of T2DM [39,41].

### 1.5. Materials and Methods

A thorough literature search was conducted to identify relevant studies for this review. The search spanned several reputable databases to ensure a wide and representative selection of articles. PubMed, Scopus, Web of Science, and Google Scholar were utilized to find additional articles not indexed in the other databases.

The literature search was carried out using a combination of keywords and Medical Subject Headings (MeSH) terms related to the topic. The key search terms included olive oil, gene expression, health effects, fatty acids, PUFA, MUFA, insulin sensitivity, and metabolic health.

Studies were selected for inclusion based on the following criteria:Inclusion Criteria: Peer-reviewed articles, including original research studies, systematic reviews, and meta-analyses; Studies published in English; Articles that focused on the effects of olive oil on gene expression, health outcomes, inflammation, and lipid metabolism; studies published in reputable scientific journals from both human and animal model research.Exclusion Criteria: Non-peer-reviewed sources such as books, conference abstracts, and opinions; Studies not focused on olive oil or rapeseed oil or those examining oils with a different composition; Articles that do not provide detailed information on gene expression or biological mechanisms related to the health effects of the oils; Studies that did not employ reliable methods for evaluating the biological or health outcomes associated with these oils.

#### Study Selection and Data Extraction

Following the initial database search, the identified articles were screened for relevance by reviewing their titles and abstracts. Then, full-text articles were assessed to ensure they met the inclusion criteria. Data were extracted from the selected articles, focusing on the following key information: study design and sample size.

Type of olive or rapeseed oil used (e.g., extra virgin olive oil, cold-pressed); specific bioactive compounds analyzed (e.g., polyphenols, fatty acid profiles); methodologies used to assess gene expression or health effects; main outcomes related to gene expression, inflammation, and metabolic health; in cases where studies presented overlapping data or duplicated results, the most comprehensive and recent data were retained.

### 1.6. Methodological Limitations

Different limitations should be considered when interpreting the results of the studies reviewed in this manuscript:In vitro vs. in vivo models:A significant portion of the studies reviewed employed in vitro (cell culture) models, which, while useful for understanding molecular mechanisms at a cellular level, do not fully replicate the complexity of human physiology. Therefore, findings from in vitro studies should be extrapolated to human health outcomes with caution. Additionally, in vivo studies (in animal models) were sometimes conducted under controlled conditions that do not entirely reflect typical human dietary patterns or environmental factors.Variability in olive oil composition:The health effects of olive oil are influenced by their unique compositions, which can vary significantly depending on factors such as geographical origin, olive cultivar, and production methods.Methodological differences across studies:Differences in the experimental designs across the studies reviewed also pose limitations. These include variations in the dosage and duration of olive oil and rapeseed oil intervention, the population (e.g., healthy individuals versus those with metabolic disorders), and the methods used to assess gene expression (e.g., RNA sequencing versus qPCR). Such methodological heterogeneity can lead to conflicting or inconsistent results, further complicating the interpretation of the overall effects of these oils on human health.

## 2. Bioactive Components of Olive Oil and Their Role in the Prevention and Management of Type 2 Diabetes Mellitus

Olive oil, a fundamental component of the Mediterranean diet, has garnered considerable interest due to its potential health benefits. Research has established associations between olive oil consumption and enhanced cardiovascular health, as well as a reduced risk of developing T2DM [42,43]. Nevertheless, the precise mechanisms through which olive oil exerts these beneficial effects remain under investigation. Olive oil, derived from the fruit of the olive tree (Olea europaea), is celebrated for its rich composition of bioactive compounds that contribute to its health-promoting properties [14]. The composition of olive oil is predominantly triglycerides (97–99%), along with a variety of minor components (1–3%) that play crucial roles in its biological and sensory characteristics [44,45].

The lipid profile of olive oil is largely characterized by monounsaturated fatty acids (MUFAs), with oleic acid being the most abundant, comprising 65–83% of total fatty acids [44]. Oleic acid is the most prevalent MUFA in human physiology [46]. Research indicates that oleic acid can positively influence metabolic health due to its anti-inflammatory and antioxidant effects. It has been shown to reduce DNA damage, promote insulin secretion, and enhance insulin sensitivity [46]. Clinical studies have reported that diets high in MUFAs are effective in lowering HbA1c levels in individuals with diabetes, thereby supporting their incorporation into dietary regimens for the management of T2DM [47].

Polyunsaturated fatty acids (PUFAs) in olive oil, particularly linoleic acid (C18:2) and α-linolenic acid (C18:3), are present in smaller proportions, typically accounting for 4–20% of total fatty acids [14]. Linoleic acid is recognized for its role in reducing low-density lipoprotein (LDL) cholesterol levels. The balance between PUFAs and MUFAs contributes to the oxidative stability of olive oil, particularly in terms of thermal degradation and the formation of volatile aldehydes, making it a suitable option for culinary applications such as frying [44].

Saturated fatty acids (SFAs) (<9.0 g/100 g) in olive oil, including palmitic acid (C16:0) and stearic acid (C18:0), constitute a minor fraction, generally around 14% of total fatty acids [14]. The favorable ratio of polyunsaturated fatty acids (PUFAs) to monounsaturated fatty acids (MUFAs), combined with the relatively low content of saturated fatty acids (SFAs)—which are linked to adverse health effects when consumed in excess—highlights olive oil’s status as one of the healthiest vegetable oils [44].

Olive oil’s minor components consist of a diverse range of bioactive compounds, which make up 1–3% of its composition. The levels and types of these compounds can vary depending on factors such as genetic characteristics, environmental conditions, fruit ripeness, harvest timing, agricultural practices, and production methods [44,48]. Tocopherols (vitamin E) are significant lipid-soluble antioxidants found in olive oil, with alpha-tocopherol (10.2–208 mg/kg) being the predominant form. It has been shown to protect cellular components from oxidative and inflammatory processes associated with aging and various degenerative diseases, including cancer [49,50].

Phenolic compounds (213–450 mg/kg) are particularly abundant in olive oil, with over 30 distinct compounds identified, making them a valuable aspect of its composition due to their robust antioxidant, antimicrobial, and anti-inflammatory properties [44,51]. These compounds include phenolic acids (e.g., caffeic acid, *p*-coumaric acid, ferulic acid), secoiridoids (such as oleuropein and its derivatives tyrosol and hydroxytyrosol), flavonoids (e.g., apigenin and luteolin), and lignans (e.g., acetoxypinoresinol and pinoresinol).

Secoiridoids, unique to olive oil and rarely found in other plant species, are abundant and have been shown to promote weight loss, improve fasting glucose levels, and enhance the body’s inflammatory and oxidative status [52]. Hydroxytyrosol and tyrosol, the most prevalent phenolic alcohols in olive oil, exhibit notable health benefits. Hydroxytyrosol, which results from the hydrolysis of oleuropein during olive maturation, has demonstrated antioxidant, anti-inflammatory, anticancer, and antidiabetic properties. Its protective effects against oxidative stress, inflammation, hyperglycemia, and hyperlipidemia have been validated in various animal models of T2DM, whether chemically, genetically, or dietarily induced [53].

Oleic acid (MUFA) is the most quantitatively prevalent fatty acid in EVOO, followed by palmitic acid (SFA) and linoleic acid (PUFA). The most abundant micronutrients in EVOO are hydroxytyrosol and oleuropein (polyphenols), followed by vitamin E (alpha-tocopherol). These compounds, oleic acid, hydroxytyrosol, oleuropein, and vitamin E are primarily responsible for EVOO’s health benefits, including insulin sensitivity, cardiovascular protection, and anti-inflammatory effects [54].

Recent data suggest that oleic acid may impact epigenetic processes, which could lead to new research into treatments based on these processes [55]. By controlling the expression of microRNA, OA can have positive anti-inflammatory effects. DNA hypomethylation was elicited in THP-1 monocytes treated with OA [56]. The hypomethylation brought on by OA improved the inflammatory profile. According to Schuldt et al., histone 3 lysine acetylation linked to elevated levels of anti-inflammatory cytokines mediates the anti-inflammatory effects of OA in fibroblasts [57]. These findings imply that histone acetyltransferase activation and SIRT-1 independence may be two distinct mechanisms through which OA effects are mediated. In monocytes, macrophages, and LPS-activated macrophages, the regulatory functions of let-7b and 155-3p in the expression of genes linked to inflammation were examined, and the possible modulatory functions of various fatty acids, including OA, were examined [58]. Activated and OA-incubated macrophages had increased levels of Let-7b. Similar outcomes in CACO cells have been reported [59].

The objective of this literature review is to analyze and synthesize existing research findings regarding the role of olive oil in modulating gene expression associated with T2DM. This review specifically aims to explore the molecular mechanisms through which components of olive oil may influence the activity of genes related to insulin sensitivity, beta cell function, glucose and lipid metabolism, and inflammatory processes linked to T2DM development. By elucidating these mechanisms, this review seeks to provide a comprehensive understanding of how olive oil may contribute to the prevention and management of T2DM, thereby offering insights into potential dietary strategies for alleviating the burden of this chronic disease. Firstly, we will review generally the various olive oil components and then we will concentrate on the role of some of these components in modulating genes associated with T2DM pathology.

### 2.1. Fatty Acids

Insulin resistance and poor glucose tolerance associated with central obesity may be mediated by elevated plasma-free fatty acids (FFAs). Since central adipocytes are insulin-resistant, increasing visceral triglyceride (TG) stores can lead to higher FFA delivery to the liver via the portal vein. This process may impair insulin signaling through mechanisms such as the Randle cycle, downregulation of the insulin signaling system, and/or reduced insulin availability to skeletal muscle due to changes in blood flow or insulin transport. While a direct causal link between intra-myocellular lipid accumulation and impaired glucose uptake has not been definitively established, TG buildup in muscle may hinder insulin action. However, while increased FFAs do not significantly impact insulin secretion in vivo, basal FFA levels support normal insulin secretion [60].

FAs are categorized as short-chain fatty acids (SCFAs) if they include fewer than six carbon (C) atoms, medium-chain fatty acids if they contain six to ten C atoms, long-chain fatty acids if they have ten to twenty-two C atoms, and very long-chain fatty acids if they contain more than twenty-two C atoms. A fatty acid’s number of double bonds determines whether it is an unsaturated or saturated fatty acid (SFA). Monounsaturated (MUFAs) and polyunsaturated (PUFAs) fatty acids are the two types of unsaturated fatty acids [61].

According to Vessby’s study, modifying the proportions of dietary fatty acids by reducing saturated fatty acids and increasing monounsaturated fatty acids can enhance insulin sensitivity. However, this does not affect insulin secretion. The beneficial impact of fat quality on insulin sensitivity is not observed in individuals with a high fat intake (>37% of total energy) [62].

The prevailing belief is that dietary fatty acids can change cell membrane functions. The fatty acid content of the cell membrane regulates fluidity, membrane protein incorporation, cellular activity, enzyme activity, ion permeability, receptor functions, glucose transporter, second messenger interactions, and contact with the insulin receptor. Any of these modifications can change the sensitivity of tissues and organs to insulin [63,64]. Reduced insulin sensitivity and glucose effectiveness have been linked to elevated SFA to PUFA ratios in the skeletal muscle cell membrane, which may raise the risk of type 2 diabetes [65,66,67]. Furthermore, the proportion and absolute amount of fatty acids can affect plasma concentrations, highlighting their significance in regulating the phospholipid composition of skeletal muscle [64]. Elevated plasma fatty acid concentrations have been shown in numerous studies to be a risk factor for type 2 diabetes, especially in people with poorly managed diabetes [68,69,70].

Olive oil, especially extra virgin olive oil (EVOO), is rich in monounsaturated fatty acids (oleic acid) and polyphenols, contributing to its anti-inflammatory, antioxidant, and insulin-sensitizing effects. The oil’s gene-modulating effects are primarily due to oleic acid, which activates PPAR-alpha, and the polyphenols, which influence NF-kB and Nrf2 pathways [71].

Rapeseed oil, conversely, contains more polyunsaturated fats, particularly omega-3 alpha-linolenic acid (ALA), which also activates PPAR-alpha and PPAR-gamma, improving insulin sensitivity and reducing inflammation. However, its higher omega-6 content (linoleic acid) can potentially promote inflammation if not balanced with omega-3s, although rapeseed oil generally maintains a good omega-6/omega-3 ratio [72]. Both oils have distinct profiles and health benefits. Olive oil is particularly beneficial for cardiovascular health, insulin sensitivity, and reducing oxidative stress. In contrast, rapeseed oil provides a valuable source of omega-3 fatty acids, contributing to vascular health and anti-inflammatory effects [73].

### 2.2. Oleic Acid (C18:1)

The two most notable dietary sources of monounsaturated fatty acids MUFAs are rapeseed oil and olive oil, which have high concentrations of erucic acid (C22:1n-9) and oleic acid (C18:1n-9). By reducing oxidative stress and glucolipotoxicity, oleic acid is essential in preventing insulin resistance and developing type 2 diabetes. Additionally, it improves the function of the hypothalamus, endothelial cells, and β-cells [74]. Oleic acid improves insulin sensitivity by raising adiponectin levels and upregulating genes. It may increase inflammatory mediators, affect glucose transport to muscles, and promote fatty acid oxidation (by raising carnitine palmitoyltransferase 145 levels) [61,75]. Consuming two to three tablespoons of extra virgin olive oil daily, which provides about 12 to 30 g of oleic acid, is a healthy and efficient strategy for incorporating it into the diet. This quantity offers additional health benefits from EVOO and promotes cardiovascular health [9]. When skeletal muscle cells are exposed to palmitic acid, oleic acid decreases the rise in mitochondrial ROS generation and the resulting suppression of insulin signaling [76].

By lowering the glycemic load and insulin demand, diets high in MUFAs increase insulin sensitivity. MUFAs improve blood lipid profiles, insulin sensitivity, and glucose regulation, which prevents β cells from dying [77]. Some of their impacts, though, are not entirely known. Despite MUFAs’ positive effects, no evidence links their consumption to a lower prevalence of type 2 diabetes [64]. Furthermore, elevated plasma MUFA levels may harm type 2 diabetes incidence [78]. Evidence shows that different MUFAs may have varied metabolic consequences.

It was discovered that both Palmitoleic Acid (POA) (0.6–3.2%) and oleic acid modulate insulin sensitivity, albeit through different mechanisms, by raising circulating levels of adiponectin and omentin and lowering inflammation markers in a study that looked at the effects of these substances on lipid metabolism, inflammation, and insulin sensitivity in a male rat model that is not obese but has prediabetic hereditary hypertriglyceridemia. The oleic acid group had significantly higher leptin levels, whereas the POA group had higher levels of adiponectin and omentin. Furthermore, it was discovered that the POA group had higher nonesterified fatty acids (NEFAs). It has been explained that NEFAs rise in response to lipolysis, which is linked to increased adipose tissue metabolic activity and helps to reduce visceral adiposity [77].

Numerous studies have shown that oleic acid and metformin have comparable protective qualities and can successfully offset palmitic acid’s harmful effects. This implies that oleic acid might have properties similar to metformin [61].

### 2.3. Palmitoleic Acid (C18:1)

Although there is no specific recommended daily intake for palmitoleic acid alone, the amount you would consume through extra virgin olive oil is relatively small. Typically, palmitoleic acid makes up about 0.3% to 1.5% of the total fatty acids in EVOO, depending on the olive variety and oil processing [79]. Diets high in palmitoleic acid improve the circulating lipid profile in both animal models and human patients, and it inhibits beta cell apoptosis brought on by glucose or saturated fatty acids [80,81,82].

In spontaneously diabetic KK-Ay mice, palmitoleic acid favors insulin resistance, hyperglycemia, and hypertriglyceridemia. Insulin resistance is the primary underlying metabolic disturbance, and other risk factors for type 2 diabetes mellitus include hyperglycemia, dyslipidemia, insulin resistance, and metabolic syndrome. Reduced responsiveness of target tissues (liver, skeletal muscle, and adipocytes) to normal circulating levels of insulin, followed by a gradual decrease in pancreatic insulin production, are the hallmarks of insulin resistance [83]. In skeletal muscle, free fatty acids—specifically, saturated fatty acids—decrease glucose consumption and increase insulin resistance [84]. Nonetheless, there are signs that monounsaturated palmitoleic acid enhances the mechanism for glucose transfer into skeletal muscle cells in mice and improves glycemic control [85]. The glucose transporters GLUT1 and GLUT4 are upregulated, which at least partially mediates this impact [86,87]. Palmitoleic acid, a triglyceride, has been demonstrated in hyperinsulinemic-euglycemic clamp tests to enhance the insulin signaling pathway in mice [85].

Several studies have demonstrated the strong correlation between adipocytokines, mainly produced from adipose tissue, and type 2 diabetes associated with insulin resistance [88,89]. One important adipocytokine, adiponectin, has been demonstrated to improve insulin sensitivity, at least in part, by reducing hepatic glucose production and promoting β-oxidation in skeletal muscle [90]. However, in different investigations, palmitoleic acid treatment did not alter the levels of adiponectin mRNA expression, indicating a minimal gene expression influence of palmitoleic acid on adiponectin. Conversely, adipocytokines TNFα and resistin, which have been shown to contribute to insulin resistance, had their mRNA levels downregulated by repeated palmitoleic acid delivery [91].

Therefore, it is hypothesized that palmitoleic acid’s favorable effect on improving insulin resistance may partly be due to its suppression of pro-inflammatory gene expression. Furthermore, some findings demonstrate that repeated administration of palmitoleic acid increased pancreatic weight. The pancreas is unable to produce enough insulin in people with type 2 diabetes, and there is evidence that this impairment is exacerbated by beta cell loss [92,93].

### 2.4. Stearic Acid

Stearic acid (1.4–3.0%) has an 18-carbon chain and is a saturated monobasic acid. It is produced by hydrogenating vegetable or cottonseed oil or hydrolyzing animal fat. Cellular malfunction brought on by excess nonesterified fatty acids (NEFA) is known as lipotoxicity. Research suggests that lipotoxicity is key in developing type 2 diabetes and contributes to beta cell damage [94].

According to in vitro studies, increasing stearic acid causes significant lipotoxicity in hepatocytes and skeletal muscle cells [95,96]. Insulin resistance is closely associated with a higher percentage of stearic acid in the blood [97]. Therefore, it is thought that increasing the blood’s stearic acid content contributes most to beta cell lipotoxicity compared to other NEFAs and is a major factor in the development of type 2 diabetes. It is clear that increased endoplasmic reticulum (ER) stress plays a significant role in lipotoxicity caused by transcriptional reprogramming, activation of c-Jun *N*-terminal kinase, CCAAT/enhancer binding protein homologous protein (CHOP), p53, and mitochondrial apoptosis pathways, even though the molecular mechanisms of stearic acid-induced lipotoxicity are not entirely understood [98,99,100]. MicroRNAs (miRNAs) have become essential regulators of hepatocyte and cardiac muscle lipotoxicity due to their role in post-transcriptional control under ER stress [101,102]. However, how miRNA contributes to stearic acid-mediated lipotoxicity in islet beta cells is unknown.

A study conducted by Lu et al. (2016) has demonstrated that pancreatic beta cells experience severe lipotoxicity when exposed to high levels of circulating stearic acid. Stearic acid cytotoxicity is largely mediated by the overexpression of miR-34a-5p and the corresponding suppression of BCL-2/BCL-W. Additionally, it has been proposed that PERK regulates the production of miR-34a-5p, and p53 mediates the upregulation of miR-34a-5p brought on by PERK [103].

Remarkably, the findings of the Tsuchiya study and colleagues [104] demonstrated that saturated FFA stearic acid functions as a potent inhibitor of the negative regulator of insulin signaling, protein tyrosine phosphatase 1B (PTP1B) [104,105]. This may improve insulin receptor signaling by promoting glucose absorption into adipocytes and inhibiting tyrosine dephosphorylation of the receptor.

Notably, the ratio of stearic acid to palmitic acid was inversely correlated with type 2 diabetes, suggesting that a lower 5-year risk of type 2 diabetes was linked to higher expression of the Elovl6 enzyme. Conversely, a higher 5-year risk of developing type 2 diabetes was related to the ratios of palmitoleic acid to palmitic acid and oleic acid to stearic acid, which indicate elevated expression of stearoyl CoA desaturase-1 (SCD1). These ratios remained somewhat attenuated despite further correction for acute insulin response (AIR) and insulin sensitivity [106].

### 2.5. Palmitic Acid (C16:0)

Significant metabolic abnormalities are frequently seen in patients with type 2 diabetes, which raises the risk of cardiovascular complications. Changes in fatty acid metabolism, namely the buildup of saturated fatty acids (SFAs), are crucial among these disruptions [107]. The most prevalent saturated fatty acid in the blood, palmitic acid (PA, C16:0), has been directly linked to severe adverse cardiovascular events and plaque susceptibility in type 2 diabetes [108]. There is no specific “recommended dose” of palmitic acid alone. However, it is essential to consider palmitic acid as part of EVOO’s overall fatty acid profile. Typically, palmitic acid represents 7% to 15% of the total fat content in extra virgin olive oil. Two to three tablespoons (about 20 to 30 mL) of extra virgin olive oil provide approximately 1.4 to 4.5 g of palmitic acid, assuming that 7 to 15% of the oil’s fat content is palmitic acid [109]. Therefore, palmitic acid’s effect on atherosclerotic plaque instability highlights, at least in part, the link between palmitic acid and elevated cardiovascular risk in T2DM patients. Some cardiovascular research has focused on the complex connection between atherosclerotic plaque vulnerability and type 2 diabetes. The results highlight the increased risk of plaque rupture in individuals with type 2 diabetes, which is caused mainly by the fibrous cap becoming thinner. Numerous investigations have shown elevated serum palmitic acid concentrations in T2DM patients, underscoring the dysregulation of lipid metabolism in diabetes [110]. Some findings in the murine model suggest that the resulting rise in palmitic acid levels plays a significant role in the establishment of susceptible plaques.

A common SFA in the bloodstream, palmitic acid, has been connected to the development and advancement of coronary artery disease and atherosclerosis [111]. According to specific research, people with type 2 diabetes have noticeably higher levels of palmitic acid than people without the disease. Interestingly, there was a strong correlation between this elevation and increased susceptibility to atherosclerotic plaque. Examining palmitic acid’s molecular function reveals that it plays a crucial role in controlling macrophage dynamics by affecting their polarization and activation [112]. It has been demonstrated that palmitic acid specifically tilts macrophages toward the pro-inflammatory M1 phenotype, increasing inflammation and, in turn, plaque vulnerability [113]. Palmitic acid’s stimulation of the TLR4 receptor is a noteworthy biochemical mechanism behind this action [114].

### 2.6. α-Linolenic Acid

Because α-Linolenic acid (ALA) has been linked to enhanced insulin sensitivity and a decreased risk of type 2 diabetes, there is evidence that it helps glycemic control [115,116]. Finding the physiological effects of ALA (0.46–0.69%), the only plant-derived omega-3 accessible, is critically needed, given recent research demonstrating its involvement in managing type 2 diabetes and the mounting strain on fish fisheries worldwide [117].

Those who are at risk for coronary heart disease may benefit from consuming 1.5 to 3 g of ALA daily, according to the American Heart Association [118]. Even though several health organizations recommend eating more foods high in ALA, its benefits for type 2 diabetes are not fully understood. Through possible roles in gene regulation, fat metabolism, and adipocyte development, ALA has been shown in both in vitro and animal studies to influence insulin sensitivity and regulate glucose homeostasis [119]. Studies on humans have revealed conflicting results about how dietary ALA affects glycemic indices in healthy and type 2 diabetic populations [120].

According to preclinical research, PPARγ and other widely dispersed peroxisome proliferation-activated receptors are activated by omega-3 PUFAs, such as ALA [121]. In T2DM, PPARγ activation may result in notable improvements in whole-body insulin sensitivity and, consequently, in glycemic control metrics [122]. There may be a relationship between alpha-linolenic acid (ALA) consumption and improved whole-body insulin sensitivity. Adipose tissue plays a key role in modulating the effects of peroxisome proliferator-activated receptor gamma (PPARγ) on insulin sensitivity. ALA, the most prevalent omega-3 fatty acid in adipose tissue, may influence PPARγ activity, thereby enhancing insulin sensitivity and contributing to better metabolic health [123].

The acceptable upper intake level (UL) for ALA was not determined. Despite the absence of toxicological data on the substance, few side effects have been documented thus far [124]. The risk of prostate cancer and ALA, however, may be a worry. According to Brouwer et al.’s systematic review and meta-analysis of prospective cohorts, ALA induced a higher risk of prostate cancer but a lower risk of fatal heart disease. On the other hand, another meta-analysis of case-control and prospective cohort studies failed to support a significant association between ALA and the risk of prostate cancer [125].

### 2.7. Linoleic Acid

The primary Ω-6 polyunsaturated fatty acid (PUFA), linoleic acid (LA), makes up 80–90% of the total dietary PUFA content [126]. The connection between LA (6.6–14.8%) and T2DM has triggered an ongoing conversation with various viewpoints and scholarly discussions. Unquestionably, a meta-analysis of prospective cohort studies shows a strong relationship between higher dietary intake of LA and higher body concentrations of LA, which considerably lowers the risk of developing type 2 diabetes [127]. In line with the findings of MR causal inference, a thorough integrated analysis encompassing 20 prospective longitudinal investigations unequivocally demonstrates that elevated LA concentration is highly linked to decreased susceptibility to T2D compared to arachidonic acid [128]. Furthermore, this study shows that elevated levels of LA can alleviate insulin and glucose resistance, which is entirely consistent with the findings of the causal inference: first, glycated hemoglobin can be utilized as a marker to evaluate the average plasma glucose levels throughout the preceding three months [129]. Second, it can be said that LA may improve insulin resistance based on the Homeostatic Model Assessment for Insulin Resistance (HOMA-IR) index, which measures insulin resistance. It is computed as fasting insulin (μU/mL) × fasting blood glucose (mmol/L)/22.5 [130]. We discovered a relationship between T2D and lower levels of LA in reverse causal inference analysis, which is in line with other findings [131].

Researchers have proposed several possibilities, but the precise causative mechanism by which LA levels lower the likelihood of acquiring T2D incidence risk has not yet been entirely determined. When incorporated into phospholipids, LA, an essential part of cell membranes, can change the membrane’s fluidity and control insulin sensitivity [132]. Elevated levels of LA can raise the concentrations of total and High Molecular Weight (HMW) adiponectin [133]. A recent study found that lipocalin levels and insulin sensitivity were positively correlated. Lipocalin can lower blood glucose and improve insulin sensitivity by upregulating IRS-2 in the liver [134,135]. This could be one way that LA lowers the chance of getting type 2 diabetes.

Furthermore, LA, present in plant oils, nuts, and seeds, is regarded as a PUFA in most Western diets [136]. Delta-6 desaturase breaks down LA in different tissues to produce gamma-linolenic acid (GLA), which quickly extends to dihomo-gamma-linolenic acid (DGLA). Delta-5 desaturase can further desaturate DGLA to produce arachidonic acid (AA). However, only a part of DGLA is transformed into AA due to the low activity of human delta-5 desaturase [137]. Prostaglandin E1 is one of the DGLA metabolites that can improve insulin action [138]. Many significant bioactive lipid mediators, including series 2 prostaglandins (such as PGE2, PGD2, PGF2α, and PGI2), are also precursors of AA [139]. PGD2 also controls insulin. Hematopoietic PGD synthase in macrophages is the primary source of PGD2 in white adipose tissue, which causes macrophages to shift from an inflammatory M1 state to an anti-inflammatory M2 state. Adipose insulin sensitivity positively correlates with this macrophage polarization [140]. According to these findings, LA metabolites PGE1 and PGE2 may increase insulin sensitivity and improve adipose insulin resistance. PGD2 may also lower the incidence of type 2 diabetes by controlling macrophage polarization.

Although the exact mechanism by which T2D lowers LA levels is unknown, one potential explanation is a marked increase in Lactobacillus abundance in T2D patients’ gut microbiomes. Lactobacillus encourages the transformation of LA in the gastrointestinal tract, which lowers circulating levels of LA and the amount of LA absorbed by the gastrointestinal tract. This phenomenon could explain the decreased LA concentration in T2D patients [141].

### 2.8. Carotenoids

Carotenoids (0.53–31.51 mg/kg) are a class of naturally occurring antioxidants found in fruits, vegetables, and marine life as vibrant hues. Olive oils contain a relatively rich variety of carotenoids (i.e., β-carotene, lutein, violaxanthin, neoxanthin, and other xanthophylls molecules responsible for providing the color of these foods [142]. Additionally, a negative correlation was identified between fasting blood insulin levels and metabolic syndrome and serum carotenoid (lycopene, lutein, and β-carotene) concentrations [143]. Carotenoids have been shown in numerous studies to lower the incidence of type 2 diabetes in both men and women [144]. Additionally, a negative correlation between carotenoid intake and Hemoglobin A1C (HbA1c) level has been noted [145]. Furthermore, research has demonstrated that carotenoids, including lutein, zeaxanthin, and lycopene, protect against diabetic retinopathy [146].

Numerous research studies have examined the role of carotenoids in preventing and treating type 2 diabetes due to their potent antioxidant properties. For instance, research by Sugiura et al. revealed that serum levels of α-carotene and β-cryptoxanthin are linked to a decreased incidence of type 2 diabetes in middle-aged and older Japanese patients [147]. The association between serum levels of five carotenoids and the risk of type 2 diabetes during 15 years was assessed in another study [148]. The findings showed that, in non-smoking patients but not in smokers, there was an inverse relationship between serum carotenoids and type 2 diabetes. Ylonen et al. similarly reported such favorable outcomes. They demonstrated that diabetes patients had considerably decreased serum concentrations of lutein, zeaxanthin, lycopene, α-carotene, and β-carotene [149].

### 2.9. Squalene

Olive oil and shark liver are natural aliments containing different bioactive compounds, including squalene, a triterpene hydrocarbon that occurs naturally.

Squalene (200–8260 mg/kg) emulsion administered intravenously demonstrated safety and good tolerance in clinical and preclinical investigations. However, it was cleared from the bloodstream more slowly than plant sterols and triglycerides [150,151]. This substance also showed a preferable, safe, and effective vaccination adjuvant [152,153]. Squalene was administered to high-fatty-diet (HFD) rats for only two weeks in the current investigation; yet, during that period, it drastically lowered Blood Glucose Level (BGL) levels but not leptin levels. This effect will probably be more noticeable over a more extended study period. High serum squalene levels have been associated with visceral obesity, according to Petlola et al. [154]. They also suggested that squalene in adipose tissue has detrimental effects on abdominal fat, making it one of the markers of metabolic syndrome.

Additionally, the sarcoplasmic reticulum releases too much calcium under hyperglycemic circumstances, which triggers apoptosis. During hyperglycemia, the overproduction of reactive oxygen species (ROS) triggers the release of pro-inflammatory cytokines such as IL-6, IL-1β, TNF-alpha, and IFN-γ. These inflammatory molecules contribute to the development of insulin resistance and the progression of T2DM by further impairing glucose metabolism and exacerbating cellular stress. Serious harm to the beta cells of the pancreas may ensue from this, which would prevent insulin synthesis. Squalene lowers the lipid peroxidation marker (MDA) and raises antioxidant status, including SOD, GST, GPx, CAT, and GSH, according to earlier research [155]. Thus, in vivo studies demonstrate the antidiabetic and antioxidant qualities of squalene.

### 2.10. Triterpenoids

Oleanolic acid and other related triterpenoids derived from olives influence various signaling pathways, demonstrating a broad spectrum of pharmacological effects, particularly in the context of inflammation [156]. In clinical studies, Oleanolic acid doses typically range from 50 mg to 100 mg per day in concentrated supplement forms. These doses are far higher than what is obtained from EVOO. Since the Oleanolic acid content in EVOO is relatively low, a typical daily intake of 1 to 2 tablespoons (15–30 mL) of EVOO is considered safe and beneficial for overall health, providing minimal doses EVOO’s Oleanolic acid. The oleanolic acid from this level of EVOO consumption is about 3–9 mg/day, which is below the higher dosages used in clinical trials [157].

Diabetes mellitus is one of the metabolic disorders for which some medicinal plants and their derivatives, such as triterpenes, continue to show significant ameliorative effects [158,159]. The beneficial effects of lanosterol triterpenes from *Protorhus longifolia*, for instance, against several diabetes-related problems, such as pancreatic β-cell destruction, have been gradually documented by our group [160,161]. Evidence of the beneficial effects of triterpenes on issues associated with diabetes has grown over time [162,163]. Remarkably, many natural substances, including some triterpenoids, work therapeutically by activating AMPK [164]. Therefore, these substances may be crucial for β-cell survival and reducing insulin resistance in diabetics. Thus, there is currently a great interest in better understanding how natural products might treat PI3-K/AKT or AMPK activation, including downstream consequences such as oxidative stress and pro-inflammatory cytokines in β-cells.

Also, triterpenoids function as inhibitors of protein tyrosine phosphatase 1B (PTP 1B). Protein tyrosine phosphatases (PTPases) control metabolic and cellular signaling pathways [165]. The two main categories of PTPases are intracellular and transmembrane. PTP 1B is a member of a class of intracellular enzymes that negatively regulate the leptin signaling system and insulin receptors, for this reason, they are more involved in diabetes control. It controls the receptor β-subunit’s dephosphorylation mechanism [166]. PTP 1B inhibitors may help reduce insulin resistance and return insulin and plasma glucose levels to normal without causing hypoglycemia [165].

Triterpenoids affect glycolytic and associated enzymes. Using NADH as a reducing equivalent, glycerol-3-phosphate dehydrogenase (G3PDH) catalyzes the reversible reduction of glycerone phosphate to produce glycerol-3-phosphate, a crucial step in lipid metabolism. However, the function of glyceraldehyde-3-phosphate dehydrogenase (GAPDH), a crucial enzyme in glycolysis, extends beyond its role in the metabolism of carbohydrates. GAPDH’s multifunctionality in cellular processes beyond energy production is further demonstrated by its involvement in membrane fusion, phosphotransferase activity, and apoptosis [167].

Gymnemic acid, a combination of roughly ten oleanane-type triterpene saponins present in Gymnema Sylvestre (Asclepiadaceae) leaves, has been shown to impact these enzymes.

### 2.11. Phytosterols

As an outcome of the implications of chronic hyperglycemia, human tissues persist in an environment of stress due to decreased glucose consumption by body cells, increased fat mobilization from fat storage cells, and protein depletion. Furthermore, by supporting healthy physiological metabolism and cellular activity, functional nutrients like phytosterols aid in the body’s recovery from these crises. Along with phytosterol derivatives, they are plant-derived steroid molecules that are abundant in nature and share structural and functional similarities with cholesterol, which can be found in vegetables, grains, nuts, wood pulp, legumes, cereals, and interestingly in olive oil [168]. One tablespoon (15 mL) of extra virgin olive oil provides 7.5 to 30 mg of phytosterols, or two tablespoons provide 15 to 60 mg. For therapeutic effects (e.g., lowering cholesterol), the typical dose of phytosterols is 1.5–3 g per day. Supplementation or consuming phytosterol-enriched foods would be required, as EVOO alone provides much lower amounts [169].

Sitosterol, stigmasterol, and campesterol are the most abundant phytosterols in the human diet. They can be found in free form, as fatty acid/cinnamic acid esters, or as glycosides broken down by pancreatic enzymes. There is growing evidence that phytosterols and diets containing them help regulate insulin resistance, lipid and glucose metabolism, and more [170,171]. Nevertheless, few published studies have examined the benefits of sterol management in treating diabetes.

According to our earlier findings, β-sitosterol lowers plasma glucose levels through either one of two mechanisms or both: either gut glucose absorption is decreased, or the glycogenic and glycolytic pathways are increased. In contrast, the gluconeogenesis and glycogenolysis pathways are decreased. The metabolism of glucose may be impacted by this [172]. Similarly, several investigations have demonstrated the antidiabetic effects of β-sitosterol [173,174].

Moreover, the livers of mammals can convert phytosterols into C21 bile acids instead of the typical C24 bile acids. Phytosterols (approximately 300 mg/100 g) have been shown to have anti-inflammatory properties, lower cholesterol, lessen the risk of diabetes and coronary heart disease, activate apoptosis in cancer cells, and aid in the prevention and treatment of disease in both humans and animals [175]. Studies conducted both in vitro and in vivo showed a notable change in the antidiabetic effect of stigmasterol and a potential mechanism. In L6 cells, stigmasterol increased GLUT4 translocation by 1.44 times. L6 cells were given varying doses of phytosterol, significantly affecting their ability to absorb glucose. By reducing fasting blood glucose levels and blood lipid indicators, including cholesterol and triglycerides, treating SMR to KK-Ay mice improved insulin resistance and oral glucose tolerance. White adipose tissue, skeletal muscle, and L6 cells showed elevated GLUT4 expression [176]. These findings suggest that phytosterols activate insulin receptor (IR) and GLUT4 to enhance glucose regulation in the liver and other peripheral organs of people with type 2 diabetes.

### 2.12. Tocopherols (Vitamin E)

Tocopherols and sterols are minor constituents of olive oils that play a significant role in determining the quality of the oil [177]. Antioxidant supplements, like vitamin E, have recently been demonstrated to improve endothelial cell dysfunction in diabetic patients [178]. Nevertheless, whether this is mediated by other mechanisms or vitamin E’s impact on insulin resistance and glycemic indices is unclear. In contrast to the standard medications used to treat diabetes, it should be mentioned that taking supplements containing these antioxidants is less expensive and linked to fewer adverse effects.

Reduced vitamin E levels have been shown to positively correlate with risk factors for type 2 diabetes mellitus (T2DM), such as insulin resistance and hyperglycemia [179]. According to some theories, vitamin E prevents glucose oxidation, which requires protein glycosylation and the synthesis of hemoglobin A1c (HbA1c) [180].

Furthermore, a previous meta-analysis of prospective cohort studies revealed that consuming more vitamin E-rich foods like nuts, seeds, liquid oil, and raisins was linked to a lower risk of diabetes and hyperglycemia [181]. Additional data support the positive impact of following diets high in vitamin E on diabetes patients’ glycemic management [182]. Contrary to that meta-analysis, results from randomized controlled trials (RCTs) examining the impact of vitamin E supplementation on insulin resistance and glycemic control in various forms of diabetes are contradictory. While some studies from Western and Asian countries demonstrated that vitamin E supplementation improves insulin resistance and glycemic indices in T2DM patients [183,184], other studies from these regions did not report such a significant effect on patients with T2DM and diabetic nephropathy [185,186]. On the other hand, some research found that Asian individuals with type 2 diabetes experienced a notable rise in blood glucose levels after taking vitamin E supplements [187,188].

In diabetic individuals, especially those with type 2 diabetes, vitamin E consumption dramatically lowers fasting insulin, HOMA-IR, and HbA1c values. Additionally, trials with an intervention length of fewer than 10 weeks indicated that vitamin E supplementation significantly reduced fasting blood glucose. Furthermore, researchers discovered that 400–700 mg of vitamin E daily is the ideal dosage for regulating insulin and HbA1c levels. Overall, patients with type 2 diabetes are advised to take 400–700 mg of vitamin E daily, as there have been no documented adverse effects. Therefore, vitamin E can be offered as an adjunct to these patients’ primary therapy (i.e., medicines) [189]. However, since little research exists on the subject, people with type 1 diabetes and those with diabetic nephropathy and/or neuropathy should proceed cautiously with this suggestion. More RCTs, especially those with a low risk of bias, are needed to evaluate the impact of vitamin E supplementation on the biochemical parameters of individuals with these conditions [190].

### 2.13. Phenolic Compounds

One of the most prevalent secondary plant metabolites in the plant kingdom is polyphenols, also known as phenolic chemicals. In its structure, one or more aromatic hydrocarbons have one or more hydroxyl groups directly connected to them. The fundamental representative phenol is used to name the entire group. Flavonoids and non-flavonoids are the two primary categories of phenolic chemicals with approximately 8000 structural variations.

Insulin resistance frequently leads to the development of T2DM in people with metabolic syndrome (MS). As a result, it is claimed that T2DM is predicted by MS and that T2DM development is five times more common in MS patients [191,192]. Polyphenols found in olive oil positively affect MS and, in turn, T2DM [193]. Other than oleic acid, some of the polyphenols in olive oil also benefit human health [6]. The reduction of T2DM is positively connected to an adequate daily intake of polyphenols [194,195].

Polyphenols influence cellular signaling pathways that regulate inflammation and oxidative stress. Key transcription factors, such as nuclear factor erythroid 2-related factor 2 (Nrf2), play a critical role in governing antioxidant defenses and associated enzymes. By activating these enzymes, polyphenols enhance the body’s intrinsic antioxidant capacity through the modulation of these pathways [196]. The redox-activated transcription factor Nrf2 plays a central role in hormetic dose responses as well by upregulating endogenous antioxidant and anti-inflammatory adaptive mechanisms. Nrf2 is activated by various oxidative stressors, including exercise, caloric restriction, chemicals, radiation, and aging-related stress. It interacts with other transcription factors to enhance metabolic resilience and adaptive homeostasis. This mechanism highlights the evolutionary conservation and efficiency of Nrf2 in protecting biological systems through modest, resource-conserving responses. Nrf2’s role as a hormetic mediator offers insights into resilience, detoxification processes, therapeutic development, and inter-individual variability in responses to toxins, treatments, and age-related diseases like type II diabetes [197].

### 2.14. Flavanoids (Apigenin, Luteolin)

Due to its nutritional advantages in metabolic illnesses, such as T2DM, olive oil is a staple of the Mediterranean diet and is valued globally. In pancreatic beta cells treated with STZ, 2-deoxy-D-ribose, or cytokines, apigenin reduced pancreatic beta cell damage by enhancing cellular antioxidant defense [198,199,200]. The adverse effects of some PCs on apoptosis (caffeic acid, vanillic acid, and vanillin) and the glucose-stimulated insulin secretion (GSIS) (*p*-coumaric acid, ferulic acid, sinapic acid, and oleuropein) may offset the positive effects of some PCs (particularly hydroxytyrosol, tyrosol, and apigenin) on beta cell survival and function. It has demonstrated that a single chemical can have both beneficial and detrimental effects. For instance, *p*-coumaric acid, ferulic acid, sinapic acid, and oleuropein raise insulin content but lower GSIS, while vanilllic acid and vanillin increase GSIS but raise apoptosis levels. This might be because pancreatic beta cells are clonally heterogeneous. Indeed, it has recently been discovered that various beta cell subpopulations may be targeted differentially by insults or therapies [201].

The ability of Apigenin, the only PC that may increase all biological effects without affecting apoptosis, to phosphorylate and activate AKT and CREB, two crucial positive regulators of beta cell mass and function, has also been demonstrated [202]. Earlier research has shown that Apigenin activates AKT in various cellular systems [203]. Crucially, these protein intermediates typically act as intracellular mediators of signaling pathways mediated by tyrosine kinase receptors or G-protein-coupled receptors. Thus, it seems plausible that Apigenin functions via a particular receptor on the surface of beta cells. To clarify the mechanisms of action by which PCs affect beta cells, more research is necessary.

Luteolin (3′,4′,5,7-tetrahydroxyflavone) is a flavonoid polyphenol found in many plant groups in aglycone and glycoside forms. While EVOO is a good source of luteolin, higher doses for specific health effects would likely require supplementation with luteolin extracts. Daily, one to two tablespoons of extra virgin olive oil provide 0.15 to 1.5 mg of luteolin. Clinical studies typically use 10–100 mg per day in supplement form to experience the therapeutic benefits of luteolin [204]. Antioxidant, anti-inflammatory, antibacterial, and anticancer properties are just a few of the many biological actions that luteolin is known to exhibit [205]. Numerous experimental investigations show that luteolin has the potential to improve diabetic conditions in type 1 and type 2 diabetes models, as well as to shield mice from diabetes-related nephropathy and cognitive impairment [206,207,208]. One study of PPAR-gamma signaling shows that luteolin has both agonistic and antagonistic effects on PPAR-gamma modulation [209].

Numerous studies have demonstrated luteolin’s anti-inflammatory and antioxidant qualities, which may be necessary for kidney preservation in diabetic nephropathy (DN) patients. Zhang et al. [210] found that luteolin may reduce glomerular sclerosis and interstitial fibrosis in DN animal models by reducing oxidative stress and inflammatory response. Additionally, it might mainly decrease STAT3 activity for its biological purpose. Furthermore, Xiong et al. [211] showed that luteolin might maintain basement membrane filtration function by upregulating the expression of the Nphs2 protein and inhibiting podocyte death, fusion, and deletion in high hyperglycemia. In addition, luteolin may decrease the progression of DN by suppressing glomerulosclerosis while preserving the comparatively typical physiological structure of glomeruli [212].

### 2.15. Phenolic Acids

Research has shown that the phenolic component of olive oil regulates the expression of many genes implicated in oxidative stress and inflammation, beneficially affecting these processes [5,213]. Recently, the health benefits of olive oil have been evaluated, and it has been proposed that determining the concentration of polyphenols in olive oil is essential to establishing a cause-and-effect link [6].

Though more and more evidence points to the positive effects of a Mediterranean diet (MD) and its constituents on T2D, the precise mechanisms underlying these benefits are still poorly understood. Higher olive oil use has also been linked to a lower incidence of type 2 diabetes in a sizable portion of the US female population [214]. Although it might be challenging to distinguish the effects of EVOO from the impact of a whole diet, there is growing evidence that olive oil polyphenols are good for human health. MD significantly improves HbA1c levels [215]. The European Prospective Investigation of Cancer and Nutrition study’s findings indicate that a 1% increase in HbA1c can raise the risk of all-cause mortality by 28%, making lowering HbA1c a pertinent concern [216]. High adherence to MD has been demonstrated in both an Italian and an Australian study to dramatically improve postprandial glucose levels and HbA1c in patients with type 2 diabetes [217].

### 2.16. Lignans

One class of secondary plant metabolite with various forms is lignans [218]. Olive oil contains at least 30 phenolic compounds, the main ones being simple phenols such as tyrosol and hydroxytyrosol, along with secoiridoids and lignans. These bioactive compounds contribute to olive oil’s health benefits, including its anti-inflammatory, antioxidant, and cardiovascular protective effects [219]. Only a few numbers of very simple propenyl phenols are used by plants to produce a complex variety of secondary metabolites [220]. A notable rise in molecular complexity is a characteristic of lignan biosynthesis [220]. Plant-based foods, including seeds, whole grains, and some fruits and vegetables, are rich in lignans, which are polyphenolic compounds [221]. Lignans are classified as diphenolic chemicals because they have similar metabolic routes and are made up of two propyl-benzene units connected by a β,β′-bond [222,223]. Based on the way oxygen is incorporated and the cyclization pattern, lignans can be divided into eight structural subgroups: furan, furofuran, arylnaphthalene, aryltetralin, dibenzylbutyrolactol, dibenzocyclooctadiene, dibenzylbutyrolactone, and dibenzylbutane. Depending on the oxidation state of the lignan molecule and the types of non-propyl aromatic rings on the side chains, each subgroup can be further split [224,225].

In Western dietary patterns, lignans are the primary dietary source of phytoestrogens due to their estrogenic qualities [226]. The gut microbiota can convert the four main plant lignans—secoisolariciresinol, matairesinol, pinoresinol, and lariciresinol—into enterolignans, making them prebiotics [227,228]. According to laboratory studies on cell and animal models, lignans have antibacterial, anti-inflammatory, and antioxidant properties. Regarding antimicrobial activity, specific lignans have demonstrated antiviral and antibacterial activity, for example, against Gram-positive bacteria via changing ion channels, membrane receptors, biofilm formation, and bacterial metabolites [229]. Pinoresinol, for instance, has shown efficacy against some viruses [98].

Regarding anti-inflammatory properties, specific lignans can suppress human mast cells’ (HMC-1) NF-kB activity, a transcription factor that influences the production of inflammatory cytokines. Consequently, the production of pro-inflammatory cytokines decreased. Additionally, lignans can reduce the infiltration of inflammatory cells and inhibit the production of nitric oxide (NO) [230,231,232]. A lower risk of cardiometabolic disorders, such as type 2 diabetes, heart disease, and excessive weight gain, is linked to enterolignans, which are significant metabolites formed from the gut flora [233,234]. Due to the presence of multiple potent antioxidants, such as lignans (e.g., lariciresinol, matairesinol, secoisolariciresinol, pinoresinol, and nortrachelogenin), numerous studies have shown the powerful antioxidant activity of plant extracts [235]. Lignothane is a naturally occurring antioxidant that has an exceptionally high antioxidant effectiveness, making it a promising clinical tool for prevention and/or treatment [236].

Intake of lariciresinol was not linked to type 2 diabetes, while secoisolariciresinol showed the strongest inverse correlation among individual lignans. Different patterns of relationships with T2D may result from variations in the bioavailability of individual lignans in the human body, which may be caused by differences in the chemical structures of the lignans, their ability to bind enzymes, and their interactions with other compounds [237,238]. Under some circumstances, secoisolariciresinol may have a better bioavailability than lariciresinol, according to in vitro and animal research [239]. This may corroborate our findings that secoisolariciresinol intake was linked to an increased risk of type 2 diabetes but not lariciresinol [240].

### 2.17. Secoiridoids

Olive oil is the primary source of added fat in the Mediterranean diet and is rich in a variety of bioactive compounds that contribute to its health benefits. These include monounsaturated fatty acids, phytosterols, simple phenols (such as tyrosol and hydroxytyrosol), secoiridoids, flavonoids, and terpenoids. These compounds work synergistically to provide antioxidant, anti-inflammatory, and cardiovascular protective effects, making olive oil a key component of a healthy diet [241]. ROS plays a significant part in the pathophysiology of many diseases and the development of oxidative stress. Oxidative stress, for instance, is a key cellular characteristic in the onset of many pathological conditions, including Alzheimer’s and Parkinson’s disease, renal damage, diabetes, cardiovascular diseases, cancer, and aging. It happens when the antioxidant defense system cannot balance the excessive accumulation of ROS produced during normal cell metabolic processes [242], and it can lead to the oxidative modification of cellular macromolecules, such as lipids, proteins, and nucleic acids [243].

According to earlier epidemiological research, a Mediterranean diet is linked to a lower risk of cancer and cardiovascular disease. The nutritional benefits of the bioactive chemicals found in its main supply of fatty acids, extra virgin olive oil (EVOO), which is high in phenolic compounds, may be reflected in this [244,245]. Among other phenolic compounds, EVOO contains HTy, Ty, and their secoiridoids precursors, such as OL, OLE, or OLA. Also, consuming extra virgin olive oil instead of refined olive oil may help avoid, develop, and advance type 2 diabetes because of its high MUFA, tyrosol, secoiridoids, and lignans [246].

### 2.18. Oleuropein

Oleuropein (OLE) is a phytochemical that is a member of the secoiridoids subgroup of polyphenols. The fruit (140 mg/g) and leaves (60–90 mg/g) of the olive tree Olea europaea L. have substantial concentrations of OLE [247]. Numerous investigations have examined OLE’s capacity to guard against the detrimental consequences of a high-fat or “Western-style” diet (HFD). According to Oi-Kano et al. [248], OLE may have anti-obesogenic qualities by preventing male Sprague-Dawley rats from gaining weight when fed a high-fat diet. Increases in plasma TG, FFA, TC, and leptin levels caused by an HFD were significantly reduced by OLE treatment [248]. In rats given OLE supplements, the interscapular brown adipose tissue (iBAT) showed higher levels of UCP-1, a protein involved in adipocyte thermogenesis [248].

In another study, Jemai et al. [249] looked at how OLE affected male Wistar rats given a high-cholesterol diet (HCD) and discovered that it prevented the liver/body weight ratio from rising due to HCD. According to serum lipid analysis, OLE considerably reduced the rises in TC, TG, and LDL-C plasma levels brought on by HCD [249]. The phenolic compound’s hypolipidemic qualities were further demonstrated by its ability to raise HDL-C levels in HCD rats, which had previously decreased [249]. The detrimental effects of an HCD on the hepatic antioxidant enzymes SOD and CAT were reversed by OLE therapy [249]. Restoring ABTS scavenging ability and reducing lipid peroxidation in the liver, heart, kidneys, and aorta of HCD-fed rats supplemented with the secoiridoid were two signs that OLE supplementation enhanced antioxidant activity [249]. According to a microarray study, Kim et al. [250] gave OLE (0.03% (*w*/*w*) in diet) to mice on a high-fat diet and discovered that the hepatic gene expression of more than 90 genes had changed by more than two times. In particular, OLE therapy decreased the number of hepatic genes related to inflammation and oxidative stress. Furthermore, OLE supplementation reduced the mRNA levels of genes associated with the hepatic fatty acid absorption pathway.

The weight of brown or white adipose tissue was unaffected by OLE supplementation in HFD mice, according to Fujiwara et al. [251]. Likewise, fasting blood insulin, AUC, and blood levels of non-esterified fatty acids were unaffected by OLE. At the same time, OLE reduced HOMA-IR and fasting blood glucose levels. As demonstrated by fluorescence immunohistochemistry quantification, GLUT4 levels in muscle tissue increased after OLE administration.

### 2.19. Oleacein

A secoiridoid called oleacein (OLE) is found in olive oil and the fruit and leaves of the Olea europaea L. plant (Oleaceae). OLE can be made from oleuropein [252] or by mixing hydroxytyrosol, also called 3,4-DHPEA-EDA [253], with the dialdehydes form of decarboxymethyl elenolic acid. Studies on OLE activities have revealed that it has strong anti-inflammatory qualities. In THP-1-derived macrophages and experimental autoimmune encephalomyelitis mice, OLE reduced the levels of anti-inflammatory cytokines such as IL-10 [254,255].

Furthermore, several investigations have shown that OLE has anti-obesity effects both in vitro and in vivo. According to G. E. Lombardo et al., OLE decreased body weight and total serum cholesterol in diabetic mice and improved insulin sensitivity in animals given a regular diet [256]. OLE significantly reduced lipid accumulation in 3T3-L1 cells, decreased the levels of the proteins fatty acid synthase (FAS) and peroxisome proliferator-activated receptor gamma (PPARγ), and increased adiponectin [257]. Through the activation of the ABC transporters and the SRB1 receptor, OLE decreases lipid buildup in macrophages [258]. OLE did not alter serum triglyceride or plasma glucose levels, but it did restore the expression of the insulin-sensitive muscle/fat glucose transporter Glut-4 [256].

According to earlier research, angiogenesis and mitochondrial biogenesis are two processes necessary for preserving metabolic activity in healthy adipocytes [259]. On the other hand, several disorders, such as insulin resistance, hypertension, and atherosclerosis, can result from adipocyte malfunction caused by persistent hypoxia, inflammation, and mitochondrial dysfunction [260]. This imbalance is believed to cause type 2 diabetes mainly [261]. OLE’s ability to block specific lipid metabolism-related pathways suggests that it may be useful in managing type 2 diabetes (T2D) [262].

### 2.20. Oleocanthal

In virgin olive oil (VOO), decarboxy methyl ligstroside aglycone (also called oleocanthal) is a phenolic molecule with unique sensory and anti-inflammatory properties [263].

The biological characteristics of oleocanthal and oleacein, particularly their capacity to regulate inflammation, oxidative stress, and cell proliferation, have attracted significant scientific attention in recent years [264]. Although some animal trials have been reported, most of this information has come from in vitro research. For example, oleacein reduced adiposity, steatosis, weight gain, and insulin resistance in high-fat mice [265]. At the same time, oleocanthal has been shown to have anti-inflammatory and antioxidant effects in animal models of lupus and arthritis [266].

The nutritional therapies effectively modulated specific inflammatory markers, and it was observed a minor impact on inflammation. In particular, following both treatments, there was an increase in CXCL1, IL-12p40, and IL-1RA. Immunomodulatory molecules CXCL1 and IL-12p40 have been linked to type 2 diabetes and other inflammatory diseases [267]. They serve as chemoattractants for several immune cells, including neutrophils and macrophages, respectively. Furthermore, by competitively binding to the pro-inflammatory IL-12 receptor, IL-12p40 creates a negative feedback loop [267]. Conversely, IL-1RA is a naturally occurring anti-inflammatory protein that inhibits IL-1α and IL-1β signaling and mediates disruptions in glucose homeostasis. Given the current paucity of research on the role of this small peptide in the pathophysiology of obesity and diabetes [268], the rise in CXCL1 following olive oil treatments is perplexing; however, those of IL-12p40 and IL-1RA are consistent with an anti-inflammatory effect mediated by blocking pro-inflammatory signaling through IL-12 and IL-1 receptors, respectively [267]. Remarkably, it has also been discovered that IL-12p40 inhibits the release of IFN-γ [269], a cytokine generated during chronic inflammation and similarly linked to the processes that start obesity-induced insulin resistance and inflammation of adipose tissue [270].

It can be concluded that treatment with EVOO, which is rich in oleocanthal and oleacein, has been shown to improve oxidative and inflammatory status in individuals with prediabetes and obesity.

### 2.21. Ligstroside

The polyphenol ligstroside aglycone (LA), also known as *p*-HPEA-Elenolic acid, is involved in extra virgin olive oil (EVOO) [271]. Although there are few data on LA bioactivity, it was shown a few years ago that LA functions as an antioxidant [242]. It has also been shown to have anti-inflammatory properties by regulating and downregulating NF-κB and to have the ability to cause a state similar to calorie restriction that affects the kidney, muscle, brain, and fat tissue, primarily through the activation and the elevation of sirtuins [272].

To stop the NOD-like receptor (NLRP3) inflammasome from activating, ligstroside aglycone, a polyphenol presents in extra virgin olive oil (EVOO), interferes with and inhibits the signaling pathways of nuclear factor kappa-light-chain-enhancer of activated B cells (NF-κB), MAP kinases (MAPKs), and Janus kinase 2/signal transducer and activator of transcription 3 (JAK2/STAT3). Inflammation-related upregulation of COX-2 and microsomal prostaglandin E synthase-1 (mPGES-1) is decreased as a result. These activities demonstrate the possible anti-inflammatory qualities of ligstroside aglycone, which adds to the health advantages of EVOO [273]. It is well known that tumor necrosis factor-α (TNF-α) activates MAPKs and NF-κB, which interfere with peroxisome proliferator-activated receptor gamma (PPAR-γ) activity, decreasing insulin signaling and trigger inflammatory reactions, ultimately resulting in insulin resistance and adipose dysfunction [274]. In a separate PREDIMED substudy, the MedDiet supplemented with EVOO decreased the rate of insulin commencement in T2D patients by 12% and postponed the use of new-onset glucose-lowering drugs [275]. Furthermore, in patients with type 2 diabetes, replacing carbs with MUFAs as the primary dietary pattern improved their metabolic profile by lowering fasting plasma glucose (FPG) levels [276,277].

### 2.22. Phenolic Alcohols

Tyrosol, hydroxytyrosol (3,4-dihydroxyphenyl ethanol), and the glycoside oleuropein are the three phenolic chemicals found in the highest concentrations in olive oil. The structures of these three compounds are similar. The only structural difference between hydroxytyrosol and tyrosol is that the former has an additional hydroxy group in the meta position. The ester oleuropein is composed of hydroxytyrosol and oleic acid.

According to numerous studies, these phenols are potent in vitro inhibitors of LDL oxidation [278]. The development of atherosclerotic plaques, which are thought to play a role in the onset of coronary heart disease, is associated with the in vivo oxidation of LDL. Phenols found in olive oil have also been positively associated with processes contributing to the pathophysiology of cancer and heart disease [279].

### 2.23. Hydroxytyrosol

Olive leaves and olive oil include hydroxytyrosol (HT), a phenolic phytochemical with anti-inflammatory, anticancer, and antidiabetic effects. Creatine kinase activity and myosin heavy chain expression, which are markers of muscle cell differentiation and contraction strength, respectively, were elevated in C2C12 cells treated with HT (1–50 μM), suggesting that HT may enhance muscle adaptation to exercise [280]. Furthermore, via boosting the expression of myogenin, mitochondrial complexes I and II, and peroxisome proliferator-activated receptor gamma coactivator (PGC)-1α, HT therapy reduced the downregulation of mitochondrial biogenesis caused by tumor necrosis factor-α (TNF-α). An aqueous extract of olive pulp containing pure HT (about 1400 mg/kg mg/day for 90 days) did not cause any mortality or morbidity in mice. The HT had anti-proliferative effects on cell lines of human colon cancer [281].

To cause diabetes in male Wistar rats, Hamden et al. (2010) administered intraperitoneal injections of streptozotocin (STZ) or STZ (150 mg/kg) and nicotinamide (1000 mg/kg) [282]. Before serum and tissues from the pancreas and small intestine were separated, the rats that showed signs of mild diabetes with hyperglycemia after two weeks were given 20 mg/kg HT for two months. While HDL cholesterol rose, HT treatment markedly decreased blood glucose, plasma triglycerides, and low-density lipoprotein (LDL) cholesterol. Furthermore, HT reduced the rise in intestinal enzymes (lactase, sucrose, and maltase) brought on by STZ, frequently increasing diabetes [283].

Overall, HT protected diabetic rats from oxidative damage and had a hypoglycemic and hypolipidemic impact. Before giving HT daily by intragastric gavage in doses of either 10 or 100 mg/kg for 6 weeks, Ristagno et al. (2012) also caused diabetes in male Sprague-Dawley rats by injecting STZ (60 mg/kg) intraperitoneally [284]. In addition to preventing deficits in nerve conduction velocity (NCV), thermal nociception, and Na^+^/K^+^-ATPase activity, HT was reported to reduce the hyperglycemia-induced increases in plasma TBARS [284]. According to this study, HT may lessen diabetic peripheral neuropathy.

Acetate, hydroxytyrosol, and hydroxytyrosol glucuronides shield hemoglobin against oxidation and morphological red blood cell (RBC) alterations at 10 μM. While hydroxytyrosol glucuronides only demonstrated limited protection independent of the concentration employed, hydroxytyrosol protected RBCs from oxidative hemolysis in a dose-dependent way when H_2_O_2_ was present [285]. Ten postmenopausal women in Florence participated in a randomized crossover trial comparing high-phenol EVOO (high-EVOO; 592 mg total phenols/kg) with low-phenol EVOO (low-EVOO; 147 mg/kg). For eight weeks during each session, subjects were instructed to replace all of their regular fats and oils with the research oil (50 g/day). The comet assay was used to quantify oxidative DNA damage in peripheral blood cells obtained at each research visit. The excretion of the olive oil phenols was measured by collecting urine samples over a 24-h period. During the high-EVOO treatment, the average of the four oxidative DNA damage measures was 30% lower than during the low-EVOO treatment (*p* = 0.02). Despite the small sample size, those who consumed high-EVOO had considerably higher urinary excretions of hydroxytyrosol and its metabolite, homovanillyl alcohol. In this study, EVOO rich in phenols, especially hydroxytyrosol, was found to reduce DNA damage [286].

Moreover, following EVOO intervention, plasma hydroxytyrosol (OH-Tyr) concentration increased significantly in the elderly. It was discovered that EVOO ingestion daily improved TAC, OH-Tyr levels, and lipid profiles. Following EVOO consumption, the data also demonstrate a significant increase in catalase (CAT) in erythrocytes and a decrease (*p* < 0.05) in glutathione peroxidase (GH-PX) and superoxide dismutase (SOD) activity [287].

### 2.24. Tyrosol

One of the main phenolic compounds found in olive oil is tyrosol, 2-(4-hydroxyphenyl)-ethanol, which has been shown to have strong antioxidant properties and various physiological effects [288]. Tyrosol has recently been shown to reduce hyperglycemia in diabetic rats [289]. Nevertheless, little is known about the molecular processes behind tyrosol’s advantageous effects on diabetes.

One of the main factors contributing to the development of type 2 diabetes is pancreatic B-cell dysfunction. B-cell failure is primarily mediated by endoplasmic reticulum (ER) stress due to the metabolic syndrome’s increased insulin production. Tyrosol and tunicamycin were used to uncover pancreatic B cells, NIT-1. Tyrosol was found to reduce the dose-dependent cell death caused by tunicamycin. Tyrosol co-treatment reduced ER stress, while tunicamycin exposure elicited the unfolded protein response (UPR). Tyrosol’s actions were mediated by JNK phosphorylation.

Without influencing apoptosis levels, hydroxytyrosol, tyrosol, and apigenin improve insulin production (hydroxytyrosol, tyrosol, and apigenin), increase glucose-stimulated insulin secretion (GSIS), and promote proliferation to support beta cell health. Remarkably, a combination of tyrosol, hydroxytyrosol, and apigenin stimulates GSIS in the islets of the human pancreas [290]. Interestingly, peroxisome proliferator-activated alpha (PPAR-α) is a nuclear hormone receptor that may interact with tyrosol as a ligand.

We can conclude that the primary mediators of EVOO’s health benefits are the monounsaturated fatty acids (MUFA), particularly oleic acid. These MUFAs play a significant role in improving insulin sensitivity, reducing inflammation, and promoting cardiovascular health [291,292]. While polyunsaturated fatty acids (PUFA) and micronutrients (especially polyphenols and vitamin E) contribute to the overall health benefits of EVOO, their effects are often complementary and support the primary role of MUFAs [71].

Thus, MUFA (primarily oleic acid) is considered the most important mediator of the health-promoting effects of extra virgin olive oil, with micronutrients like polyphenols and vitamin E playing a secondary but supportive role [293].

## 3. Effects of Different Compounds on the Expression of Genes Related to T2DM

### 3.1. Effects of Oleate on the Expression of Genes Related to T2DM

Several studies have reported beneficial effects of oleate on insulin signaling by affecting the expression of PI3K signaling pathway proteins [61,294,295] In the study by López-Gómez et al., conducted on visceral adipocytes isolated from patients with morbid obesity and non-obese patients, it was demonstrated that oleate significantly upregulated the expression of IRS-1 and p110β, a catalytic subunit of PI3K, while downregulated the expression of p85α, a regulatory subunit of PI3K, and the p85α/p110β ratio in a dose-dependent manner [294]. Downregulating p85α, which has been suggested as an early molecular step in the pathogenesis of diet-induced insulin resistance, would increase the availability of the p85/p110 dimer to bind to IRS-1. This, in turn, would promote more effective activation of PI3K and enhance insulin sensitivity. This mechanism highlights how modulating the expression of p85α can improve insulin signaling and potentially counteract insulin resistance, as illustrated in Figure 2.

The liver transcriptomes of mice given high-fat diets comprising olive (O), palm (P), and hybrid palm (HP) oils were compared in one study. According to its findings, the overall quantity of fat may play a significant role in causing transcriptome alterations controlled by genes linked to FA metabolism. Curiously, the Scd1 gene was upregulated in both the P and HP groups, although cpt1a, a key participant in the oxidation of FA [296], was downregulated in the HP group. These findings imply that various oils, especially O and HP oils, cause significant alterations in the liver transcriptome. More characterization is required to determine which mRNA-level alterations are responsible for the biological consequences seen throughout the organism [297].

Some lifestyle factors likely contribute to the lower risk of cardiovascular disease in Mediterranean countries, where virgin olive oil is a significant source of dietary fat. However, one study revealed that consuming a breakfast rich in virgin olive oil and phenolic compounds can reduce the in vivo expression of several pro-inflammatory genes, thereby shifting the activity of peripheral blood mononuclear cells to a less harmful inflammatory profile [298]. According to a specific study, the development of neurodegenerative illnesses may be influenced by the downregulation of transcriptome pathways, particularly those linked to neuroinflammation, in the Traditional Mediterranean Die enriched in mixed nuts or VOO [299].

Also, microarrays were used to analyze the gene expression profiles of peripheral blood mononuclear cells (PBMNCs) from healthy persons in pooled RNA samples following three weeks of moderate and consistent VOO consumption as the primary fat source in a diet regulated for antioxidant content. qPCR was used to confirm the expression of genes. For individual samples (*n* = 10), the reaction to VOO consumption was verified using qPCR for 10 elevated genes (ADAM17, ALDH1A1, BIRC1, ERCC5, LIAS, OGT, PPARBP, TNFSF10, USP48, and XRCC5). Their potential contribution to the molecular processes underlying the onset and advancement of atherosclerosis was examined, with particular attention to a possible connection to VOO intake. According to these findings, a three-week nutritional intervention that includes VOO supplementation in amounts typical of the Mediterranean diet can change the expression of genes linked to the onset and progression of atherosclerosis [300].

The study by Alkhateeb et al. explored the impact of oleate on palmitate-induced insulin resistance in rat skeletal muscle [295]. The results of this study demonstrated that palmitate exposure significantly impaired insulin-stimulated GLUT4 translocation and the phosphorylation of AS160 and Akt-2, the key proteins in the insulin signaling pathway. Oleate administration effectively restored the insulin-stimulated translocation of GLUT4 and the phosphorylation of AS160 and Akt-2, suggesting a protective effect against palmitate-induced insulin resistance. The study indicated that this restorative effect of oleate is mediated, at least in part, by the activation of the PI3K pathway, as evidenced by the partial inhibition of these benefits by wortmannin, a PI3K inhibitor, although this effect is not dependent on 5′ AMP-activated protein kinase (AMPK) activation.

Gonçalves-de-Albuquerque et al. conducted a study using a sepsis-induced mouse model to assess the effects of oleate on lipid metabolism [301]. Their findings indicated that oleate supplementation significantly improved clinical outcomes, enhanced survival rates, and mitigated liver and kidney injury in septic mice. Specifically, oleate treatment resulted in increased expression of two essential proteins in the oxidation of fatty acids in the liver: carnitine palmitoyltransferase 1A (CPT-1A), which facilitates the transport of fatty acids into mitochondria for β-oxidation, and uncoupling protein 2 (UCP2), which modulates the proton gradient across the mitochondrial membrane, reducing mitochondrial membrane potential. Furthermore, oleate inhibited the decrease in AMPK expression, a key enzyme in energy homeostasis, increasing its expression in the liver. These combined effects resulted in diminished ROS production and enhanced fatty acid oxidation, thus decreasing levels of plasma non-esterified fatty acids.

Several studies have highlighted the anti-inflammatory properties of olive oil components [302,303]. In the study conducted by Sun et al., the effects of oleate on insulin sensitivity were examined using human hepatic cells [303]. The research revealed that oleate alleviated palmitate-induced insulin resistance by modulating inflammatory pathways associated with ROS and the JUN-encoded protein, a key inflammatory mediator that disrupts insulin signaling. Specifically, oleate was found to counteract the ROS/JUN pathway by inhibiting cellular ROS production, thereby restoring Akt phosphorylation levels that were diminished by palmitate.

In the study conducted by Miklankova et al., the researchers investigated the effects of oleate supplementation on lipid metabolism and inflammation using a prediabetic rat model [77]. The findings demonstrated that oleate supplementation led to significant alterations in arachidonic acid metabolism. Specifically, oleate was associated with reduced levels of arachidonic acid and its pro-inflammatory metabolites, including 20-HETE, in the membrane phospholipids of peripheral tissues. These results suggest that oleate exerts beneficial effects on lipid metabolism and possesses anti-inflammatory properties. This reduction in pro-inflammatory arachidonic acid-derived metabolites is associated with lower chronic inflammation. Additionally, oleate slightly increased the production of adiponectin in visceral adipose tissue, which was also shown to be beneficial for insulin sensitivity [304].

Several studies have demonstrated the beneficial effects of oleate on beta cell function, including the inhibition of the pro-apoptotic effect of saturated fatty acids, and the promotion of autophagy in beta cells [46,305,306,307,308,309,310].

In the study by Nemecz et al. [305] oleate was found to enhance beta cell function through multiple protective mechanisms. Using the human insulin-releasing beta cell line 1.1B4, the researchers demonstrated that oleate promoted neutral lipid accumulation and insulin secretion and reversed the negative effects of palmitic acid through reducing ROS production and downregulating stress-related proteins such as BiP, eIF2α, ATF6, XBP1, and CHOP, which are involved in the UPR and apoptosis. By alleviating these stress responses, oleate preserved beta cell viability and functionality, thereby enhancing insulin secretion even in the presence of palmitate.

Šrámek et al. [307] investigated the effects of oleate on beta cell function utilizing a human pancreatic β-cell line. The study’s results indicated that oleate substantially alleviates the adverse effects of stearic acid on beta cells. Specifically, oleate was found to inhibit apoptosis induced by stearic acid by preventing the activation of the p38 MAPK pathway and counteracting the inhibition of the extracellular signal-regulated kinase (ERK) pathway, which is a cellular main pro-proliferative signaling pathway. Additionally, oleate also reduced the activation of ER stress signaling pathways induced by stearic acid, including IRE1 and PERK pathways through inhibiting CHOP and BiP expression, and inhibiting the induction of XBP1 splicing.

Liu et al. [46] conducted a study using an in vitro model with the rat insulinoma cell line INS-1E to investigate the effects of oleate. The study demonstrated that oleate significantly reduces palmitate-induced ER stress, apoptosis, and inflammation. Oleate’s protective effects were evidenced by the reduced expression of ER stress markers such as CHOP and phosphorylated eIF2α (*p*-eIF2α), and a decreased ratio of XBP1 spliced form to unspliced form. Additionally, oleate increased the expression ratio of the anti-apoptotic protein Bcl-2 to pro-apoptotic protein Bax, and reduced palmitate-induced hyperexpression of cleaved caspase-3, which is the active form of caspase-3, an essential executioner protease in the process of apoptosis and a marker of cytotoxicity. In line with this study, Oberhauser et al. [310], using the murine-derived INS-1E beta cell line, demonstrated that chronic exposure to oleate, particularly under high glucose conditions, provided a stronger protective effect against glucotoxicity compared to other unsaturated fatty acids, such as linoleate (C18:2) and linolenate (C18:3), by reducing the levels of cleaved caspase-3.

Sargsyan et al. [308] conducted a study to evaluate the protective effects of oleate on beta cell function in the context of palmitate-induced toxicity, utilizing a mouse insulinoma cell line model. The study found that oleate mitigated the adverse effects of palmitate by inhibiting pro-apoptotic pathways within the ER stress response. Specifically, oleate altered the expression of several ER stress markers, including the downregulation *p*-eIF2α and CHOP.

However, certain studies have indicated that prolonged exposure to oleate may lead to impairment of beta cell function. According to the study by Plötz et al. [311], oleate exhibits significant toxic effects on human beta cell function. Using the human EndoC-βH1 beta cell line as the primary model, along with isolated rat and human islets, the researchers demonstrated that incubation with oleate resulted in increased caspase-3 activity, indicating apoptosis. The study concluded that oleate-induced lipotoxicity is comparable to that of palmitate, challenging the conventional view of oleate as a protective lipid. This finding suggests that oleate may contribute to beta cell dysfunction by inducing apoptosis.

In the study by Suzuki et al. [312], which utilized the INS-1 pancreatic beta cell line derived from rats, oleate was demonstrated to enhance mitochondrial energy metabolism. This enhancement resulted in increased ATP production and improved glucose sensitivity. This was evidenced by a faster increase in intracellular ATP concentration and sustained mitochondrial membrane potential following glucose stimulation. The mRNA expression of caspase-3 was significantly increased in oleate-treated cells, although its activity was not high. However, prolonged exposure to oleate resulted in a reduction of oxidative stress resistance, as indicated by increased intracellular ROS levels and a higher rate of apoptosis upon oxidative stress induction. The study highlights a dual role of oleate in beta cell function: while it promotes metabolic activity and glucose responsiveness in the short term, extended exposure can compromise cellular resistance to oxidative stress and increase the susceptibility to apoptosis.

In the study by Jazurek-Ciesiolka [313], oleate’s impact on pancreatic beta cell function was investigated using both in vitro and in vivo models. The in vitro experiments involved treating INS-1E rat insulinoma cells with oleate, while the in vivo component utilized male Wistar rats fed various diets, including a trioleate-enriched diet, over a period of 12 weeks. The findings revealed that in beta cells oleate significantly promoted the nuclear translocation and transcriptional activity of Forkhead box O1 (FoxO1), which is a transcription factor involved in the regulation of genes that control cell growth, proliferation, differentiation, and survival. This nuclear accumulation was achieved through decreased phosphorylation of FoxO1 and Akt, leading to enhanced β-catenin binding to FoxO1 instead of Transcription Factor 7-Like 2 (TCF7L2), a transcription factor that is part of the Wnt signaling pathway, a crucial pathway for regulating gene expression, cell behavior, proliferation, and differentiation. Consequently, oleate shifted the transcriptional balance from Wnt/β-catenin signaling to FoxO1-mediated transcription, impairing the compensatory response of beta cells to insulin resistance. The study underscores the role of oleate in modulating beta cell function, suggesting that its influence on FoxO1 activity contributes to the failure of beta cell adaptation in the context of obesity-induced insulin resistance.

### 3.2. Effects of the Phenolic Alcohols Hydroxytyrosol and Tyrosol on the Expression of Genes Related to T2DM

In a recent study by Scoditti et al. [314], the effects of hydroxytyrosol on inflammation and oxidative stress in adipocytes were investigated. The researchers demonstrated that hydroxytyrosol significantly inhibited the TNF-α-induced expression of pro-inflammatory cytokines, chemokines, and enzymes, including monocyte chemoattractant protein-1 (MCP-1), IL-6, and cyclooxygenase-2 (COX-2). Additionally, hydroxytyrosol was found to restore the expression of key metabolic regulators, such as GLUT4 and endothelial nitric oxide synthase (eNOS). Hydroxytyrosol also significantly counteracted the expression of inflammation-related microRNAs, reduced the production of ROS, and prevented the activation of the NF-κB pathway, which is known to exacerbate inflammatory responses and insulin resistance, see Figure 3.

Wang et al. [315] conducted a study using a diet-induced obesity mouse model, which demonstrated that hydroxytyrosol administration significantly improved glucose homeostasis and insulin sensitivity. This was evidenced by reductions in fasting blood glucose and insulin levels. Hydroxytyrosol was shown to modulate ER stress markers, which are critical in the development of insulin resistance, by reducing the phosphorylation of PERK, IRE1, ATF6, and JNK, thus enhancing insulin signaling pathways. Importantly, hydroxytyrosol treatment effectively reversed the high-fat diet-induced alterations in IRS-1 and Akt, restoring the phosphorylation levels of these critical molecules in the insulin signaling pathway. Specifically, it reduced the serine phosphorylation of IRS-1 and increased the phosphorylation of Akt, thereby improving insulin sensitivity. Additionally, hydroxytyrosol contributed to the reduction of pro-inflammatory markers such as TNFα and IL-1β in adipose and liver tissues.

The study by Soylu et al. [316], conducted using diabetic Wistar rats, underscored the antioxidant properties of hydroxytyrosol and its capacity to enhance insulin production. The findings revealed that hydroxytyrosol administration significantly lowered blood glucose levels in the diabetic rats and increased the expression of insulin and peroxiredoxin-6 (Prdx6) in pancreatic beta cells. Prdx6 is an antioxidant enzyme that protects beta cells from oxidative stress, a significant factor in beta cell dysfunction and T2DM progression. The study suggests that hydroxytyrosol enhances insulin secretion and provides protective effects against oxidative damage, thereby improving pancreatic function.

According to the study by Jafari-Rastegar et al. [317]., conducted on streptozotocin (STZ)-induced diabetic Wistar rats, tyrosol was shown to effectively reduce apoptotic cell count, and to restore insulin receptor protein levels and superoxide dismutase (SOD) activity in hepatic cells. These findings suggest that tyrosol’s antioxidant and anti-inflammatory properties can attenuate oxidative stress and inflammation associated with T2DM, thereby protecting liver tissues and potentially improving overall insulin sensitivity and glucose metabolism.

Tyrosol has also been shown to protect beta cells from ER stress-induced apoptosis. According to the study by Lee et al. [318], which used both in vitro and in vivo models, tyrosol significantly reduced cell death in NIT-1 insulinoma beta cells exposed to the ER stress inducer tunicamycin. This protective effect was achieved through the inhibition of key stress and apoptotic pathways, specifically by reducing the phosphorylation of JNK and decreasing the expression of ER stress markers such as BiP, phosphorylated PERK, and *p*-eIF2α. Tyrosol also suppressed the tunicamycin-induced mitochondrial apoptotic pathway by counteracting the activation of pro-apoptotic protein Bax and the inhibition of anti-apoptotic protein Bcl-2, thus decreasing cleaved caspase 3 levels. In vivo, tyrosol supplementation to mice which was fed a high-fat diet resulted in a notable amelioration of beta cell loss and improved insulin production.

### 3.3. Effects of Secoiridoids on the Expression of Genes Related to T2DM

Oleuropein (125  mg/kg of diet) has been shown to enhance insulin sensitivity and beta cell function. The study by Alkhateeb et al. [48] found that oleuropein administration significantly restored insulin-stimulated glucose transport and AS160 phosphorylation, and partially restored GLUT4 translocation, impaired by palmitate exposure. Notably, inhibition of AMPK phosphorylation prevented the oleuropein-induced improvements in insulin-stimulated glucose transport and signaling, indicating that oleuropein’s beneficial effects are mediated through an AMPK-dependent mechanism.

According to the study by Wu et al. [319], which used an in vitro model with INS-1 beta cells, oleuropein has been shown to enhance glucose-stimulated insulin secretion and protect beta cells from hIAPP-induced cytotoxicity. The study elucidates that the insulinotropic effect of oleuropein is mediated through the activation of the ERK/MAPK signaling pathway, which is a cellular pro-proliferative signaling pathway. Oleuropein’s protective role against hIAPP-induced toxicity was found to be attributed to the 3-hydroxytyrosol moiety of the molecule. The results of this study are in line with the study by Chaari [320], which also highlighted that oleuropein and its derivatives inhibit the aggregation of hIAPP. Utilizing in vitro models, the research demonstrated that oleuropein aglycone, hydroxytyrosol, and tyrosol effectively prevent the formation of amyloid fibrils by interfering with the nucleation and elongation phases of hIAPP aggregation. Notably, oleuropein aglycone emerged as the most potent inhibitor, also significantly reducing hIAPP-induced cytotoxicity in pancreatic INS-1E cells.

Other secoiridoid polyphenols such as oleocanthal and oleacein have been shown to exhibit significant anti-inflammatory properties that could have a beneficial role in the context of T2DM. Oleocanthal is a natural phenolic compound in extra virgin olive oil (EVOO) with anti-inflammatory and antioxidant properties. Although there is no standard recommended dosage for oleocanthal, studies suggest that approximately 5–10 mg of oleocanthal per day may provide health benefits with positive effects on inflammation and cardiovascular protection [321]. The study conducted by Carpi et al. [322], using an in vitro model of human SGBS adipocytes, demonstrated that these phenolic compounds effectively attenuate the inflammatory response induced by TNF-α. By reducing the expression of pro-inflammatory genes such as IL-1β, COX-2, and MCP-1, and modulating inflammation-linked miRNAs, these polyphenols inhibit the activation of the NF-κB pathway, which is implicated in the development of insulin resistance. The suppression of NF-κB activation and the improvement of peroxisome proliferator-activated receptor gamma (PPARγ) expression suggest that oleocanthal and oleacein can mitigate adipocyte dysfunction and systemic inflammation, potentially improving insulin sensitivity and glucose homeostasis.

In the study conducted by Lombardo et al. [323], utilizing a murine in vivo model, it was demonstrated that oleacein significantly mitigates the adverse metabolic effects associated with a high-fat diet, which are frequently precursors to T2DM. Oral administration of oleacein effectively prevented weight gain, reduced visceral fat accumulation, and improved insulin sensitivity, as evidenced by lower fasting glucose and insulin levels. Studies suggest that high-quality EVOO can contain between 2 and 5 mg of oleacein per 10 g of oil. Based on this content, to obtain a significant amount of oleacein, it would be recommended to consume about 2–3 tablespoons of extra virgin olive oil per day (about 20–30 mL), which could provide the appropriate amount of oleacein and other health-beneficial polyphenols [324]. Additionally, oleacein treatment resulted in the reduction of the expression of key proteins regulating hepatic lipogenesis, leading to the enhanced liver insulin sensitivity: a transcriptional activator SREBP-1 (Sterol Regulatory Element-Binding Protein 1), and its target FAS (Fatty Acid Synthase). It also led to reduced phospho-ERK expression, a condition that has also been associated with amelioration of insulin resistance in the liver.

A recent study by Wang et al. [325], employing an in vitro model with human adipose-derived stem cells, demonstrated that oleacein has considerable potential in modulating metabolic functions relevant to T2DM management. Oleacein treatment was shown to downregulate lipid metabolism-related genes and enhance glucose metabolism, suggesting an improvement in insulin sensitivity. Key pathways affected by this polyphenol included the PI3K-Akt signaling pathway and the NF-κB signaling pathway, which are critical for regulating lipid metabolism, glucose uptake, insulin signaling, and inflammation. Additionally, oleacein positively modulated the expression of genes involved in fatty acid metabolism, glucose transport (e.g., GLUT4), glycolysis, and pro-inflammatory cytokines (e.g., TNF-α, IL-6).

### 3.4. Effects of Ligstroside Aglycon on the Expression of Genes Related to T2DM

Ligstroside aglycon (approximately an amount of 18.81 mg/kg in EVOO) is another secoiridoid polyphenol that has also been shown to exhibit antioxidant and anti-inflammatory properties that may play a beneficial role in T2DM management. Utilizing an in vitro model involving murine macrophages, the study by Castejón et al. [326], demonstrated that this polyphenol effectively reduced oxidative stress markers, such as nitric oxide production, and downregulated the expression of pro-inflammatory enzymes like inducible nitric oxide synthase (iNOS) and NADPH oxidase-1 (NOX-1). Furthermore, it inhibited key inflammatory signaling pathways, including NF-κB, MAPKs, and JAK2/STAT3, and modulated the Nrf2/HO-1 antioxidant pathway, which collectively contributed to its anti-inflammatory effects.

### 3.5. Combined Effects of Oleate and Phenolic Compounds on the Expression of Genes Related to T2DM

The study by Priore et al. utilized rat C6 glioma cells to investigate the combined effects of oleate and hydroxytyrosol on lipid metabolism [327]. The results demonstrated that both olive oil components significantly inhibit the synthesis of cholesterol and fatty acids within these cells, without affecting cell viability. The inhibitory effect was more pronounced when oleate and hydroxytyrosol were administered in combination. This inhibition was associated with a reduction in the activity of key enzymes involved in lipid biosynthesis, specifically acetyl-CoA carboxylase (ACC) and 3-hydroxy-3-methylglutaryl-CoA reductase (HMGCR), which are crucial for fatty acid and cholesterol synthesis, respectively. The study found that oleate and hydroxytyrosol decrease the mRNA abundance and protein levels of these enzymes, indicating a direct regulatory effect on lipid metabolism.

The study by Scoditti et al. investigated the effects of oleate and hydroxytyrosol on the expression of adiponectin, an adipocyte-secreted hormone with anti-inflammatory and insulin-sensitizing effects, in human and murine adipocytes under pro-inflammatory conditions induced by TNF-α [314]. The study demonstrated that both oleate and hydroxytyrosol, either alone or in combination, significantly prevent the TNF-α-induced suppression of adiponectin secretion and mRNA levels. Notably, co-treatment with both olive oil compounds restored adiponectin expression additively compared to single treatments. This additive effect was associated with a significant reduction in TNF-α-stimulated JNK phosphorylation and activation, which mediates the degradation of PPARγ, a transcription factor implicated in adiponectin gene expression. By preventing JNK activation, treatment with olive oil compounds restores PPARγ expression and activity, which in turn restores adiponectin levels, see Figure 4.

### 3.6. Effect of Phenolic Compounds in Olive Oil on the Expression of Genes Related to T2DM

The phenolic compounds in olive oil to be addressed are the Flavanoids (Apigenin, luteolin, and baicalein), Phenolic Acids, and Lignans. Dietary flavonoids, which are abundant in olive oil, exhibit a wide range of pharmacological and nutritional properties.

#### 3.6.1. Effect of Flavonoids in Olive Oil on the Expression of Genes Related to T2DM

The study conducted by Miao et al. [328] explored the antidiabetic effects of three specific flavonoids—apigenin, luteolin, and baicalein—on high-glucose and dexamethasone-induced insulin-resistant (IR) human hepatic HepG2 cells. Hepatic cells are critical in regulating the metabolism of carbohydrates, lipids, amino acids, and proteins, playing a central role in glucose homeostasis through processes such as glycogenesis and gluconeogenesis. Insulin regulates glucose uptake into cells by binding to the insulin receptor, which triggers a complex intracellular signaling cascade. Insulin receptor substrates 1 and 2 (IRS-1 and IRS-2), key downstream components of the insulin receptor, activate phosphatidylinositol 3-kinase (PI3K) [329]. In collaboration with 3-phosphoinositide-dependent protein kinase-1 (PDK1), PI3K activates Akt via phosphorylation [330]. The activation of Akt leads to the phosphorylation and inactivation of glycogen synthase kinase-3β (GSK-3β), an enzyme involved in glycogen metabolism [331]. Furthermore, insulin stimulates the translocation of glucose transporter proteins, such as GLUT4 and GLUT2, to the plasma membrane, facilitating glucose uptake. This process is also regulated by Akt activation, completing the IRS-1/2/PI3K/Akt signaling pathway and the activation of GLUT4 and GLUT2 [332,333].

In the context of diabetes, characterized by elevated glucose levels, there is an increase in the production of reactive oxygen species (ROS) and advanced glycation end products (AGEs), both of which contribute significantly to the pathophysiology of diabetic complications. ROS accumulation can elevate NADH levels, which serve as substrates for mitochondrial complex I [334]. AGEs, produced by the irreversible non-enzymatic glycation of proteins, interact with the receptor for AGEs (RAGE), triggering further ROS production and contributing to cellular dysfunction [335]. Additionally, oxidative stress resulting from increased ROS can activate the phosphorylation of the p65 subunit of nuclear factor kappa B (NF-κB), thereby activating NF-κB signaling pathways. This cascade is implicated in the activation of cellular inflammasomes, which play a prominent role in the inflammatory response in diabetes [336].

The findings from Miau et al. [328] demonstrate that all three flavonoids—apigenin, luteolin, and baicalein—enhance glucose consumption and glycogen synthesis in insulin-resistant HepG2 cells. This effect is likely mediated through the activation of glucose transporter protein 4 (GLUT4) and the phosphorylation of glycogen synthase kinase-3β (GSK-3β). Additionally, these flavonoids significantly reduce the production of reactive oxygen species (ROS) and advanced glycation end products (AGEs), which are associated with the inhibition of phosphorylated NF-κB and its p65 subunit, contributing to improved insulin sensitivity and reduced inflammation. The flavonoids also partially upregulated the expression of insulin receptor substrates 1 and 2 (IRS-1 and IRS-2) as well as key components of the PI3K/Akt signaling pathway in the IR-HepG2 cells, albeit with varying effects. Additionally, the study assessed intracellular metabolic changes induced by these flavonoids. Overall, these results suggest that apigenin, luteolin, and baicalein may exert beneficial effects on insulin resistance by modulating key molecular pathways involved in glucose metabolism and oxidative stress.

#### 3.6.2. Effect of Phenolic Acids in Olive Oil on the Expression of Genes Related to T2DM

Phenolic acids are a group of phenolic compounds characterized by the presence of a single carboxylic acid group. These compounds are primarily classified into two subclasses: hydroxybenzoic acids, derived from benzoic acid, and hydroxycinnamic acids, derived from cinnamic acid. The phenolic acids most commonly detected in virgin olive oil (VOO) include protocatechuic acid, *p*- and o-coumaric acids, *p*-hydroxybenzoic acid, caffeic acid, gallic acid, cinnamic acid, vanillic acid, syringic acid, and ferulic acid [337]. Compared to other chemical classes, phenolic acids generally occur in lower concentrations in VOO. However, these compounds are recognized for their potent antioxidant properties and play a significant role in the biological and sensory characteristics of VOO [337].

Phenolic acids are secondary aromatic metabolites widely distributed throughout the plant kingdom [338]. They are naturally occurring compounds containing two distinct carbon structures: the hydroxycinnamic and hydroxybenzoic acid frameworks. Recent interest in phenolic acids has been driven by their potential protective effects, particularly through the consumption of fruits and vegetables, against diseases associated with oxidative stress, such as type 2 diabetes mellitus (T2DM). Several phenolic acids, including gallic acid, protocatechuic acid, *p*-hydroxybenzoic acid, vanillic acid, caffeic acid, syringic acid, *p*- and o-coumaric acids, ferulic acid, and cinnamic acid, have been identified and quantified in VOO, typically at concentrations below 1 mg per kilogram of olive oil.

The multitarget actions of phenolic acids, including their roles as hypoglycemic (insulin secretagogues), anti-hyperglycemic (insulin sensitizers), and adipogenic molecules, have been investigated by Becerra Sanchez et al. [339]. In their study, two distinct in vitro models were used: RINm5F cells, a cell line that secretes insulin, and 3T3-L1 adipocytes. The researchers demonstrated that phenolic acids could act as agonists for peroxisome proliferator-activated receptors (PPARs) α and γ. In RINm5F cells, phenolic acids were found to interact with G-protein-coupled receptors (GPCRs), which activate signal transduction pathways, including the inositol trisphosphate (IP3) receptor pathway. This activation resulted in an increase in intracellular calcium ([Ca^2+^]i) levels and subsequent insulin secretion. In 3T3-L1 adipocytes, phenolic acids were shown to upregulate the expression of mRNA for PPARγ and glucose transporter 4 (GLUT4). Based on chemo-informatics analysis, it is proposed that phenolic acids may function as ligands for nuclear receptors, activating the expression of PPAR and GLUT4, thereby potentially improving insulin sensitivity.

#### 3.6.3. Effect of Olive Oil Lignans on the Expression of Genes Related to T2DM

Lignans are polyphenolic compounds found abundantly in various plant-based foods, including seeds, cereals, fruits, vegetables, nuts, tea, coffee, and olive oil. These plant-derived lignans can be efficiently metabolized by the gut microbiota into enterolignans, specifically enterolactone and enterodiol, which are subsequently bioavailable and utilized within the human body [340]. Observational studies have indicated that lignan intake is associated with beneficial effects on several metabolic parameters, including lipid profile, glycemic control, blood pressure, and endothelial function [341]. Furthermore, lignans have been shown to influence gut microbiota composition and modify microbial metabolites, which play a crucial role in inhibiting NF-κB activation, thereby suppressing the expression of pro-inflammatory cytokines [342]. Additionally, enterolignans have demonstrated antioxidative properties, which may help mitigate DNA damage and reduce lipid peroxidation [343].

It is also conceivable that lignans contribute to improved glucose homeostasis by modulating the activity of key enzymes involved in plasma glucose regulation. This hypothesis is supported by findings from a feeding trial with flaxseed lignans, which reported a reduction in insulin resistance [344]. Taken together, these effects may explain the observed association between lignan intake and a reduced risk of type 2 diabetes (T2DM) [345,346].

We can conclude that different cultivars of olive oil, which vary in their chemical composition (including fatty acids, polyphenols, and other bioactive compounds), can affect gene expression differently [347]. Several studies have shown that the bioactive compounds found in olive oil, such as polyphenols (especially hydroxytyrosol, oleuropein), monounsaturated fatty acids (mainly oleic acid), and other micronutrients, can modulate gene expression related to inflammation, oxidative stress, lipid metabolism, and insulin sensitivity [293]. However, the specific effects can vary depending on the olive cultivar used to produce the oil. Therefore, olive oils from cultivars harvested early in the season or processed with care to preserve polyphenol content tend to have more pronounced anti-inflammatory and antioxidant effects. Such oils may better modulate genes related to oxidative stress and inflammation, leading to improved health outcomes [348].

Olive oils with higher concentrations of oleic acid may better influence lipid metabolism and insulin sensitivity, potentially making them more beneficial for cardiovascular health and metabolic conditions like diabetes. The variation in gene expression effects between different cultivars implies that olive oils should be selected based on the desired health benefits. For example, a polyphenol cultivar may be preferred for antioxidant and anti-inflammatory effects, while one rich in oleic acid might be more beneficial for cardiometabolic health [43] rapeseed oil.

## 4. Discussion

The effects of olive oil on type 2 diabetes (T2DM) risk and insulin sensitivity are deeply influenced by the broader dietary context in which it is consumed. When incorporated into the Mediterranean diet, which emphasizes consuming fruits, vegetables, whole grains, legumes, nuts, and fish, olive oil enhances the synergistic interactions between these health-promoting components. This combination leads to improved metabolic health, better insulin sensitivity, and a reduced risk of T2DM [349]. However, the outcomes of olive oil consumption may vary depending on the overall dietary patterns and lifestyle factors across different populations. For individuals aiming to maximize the health benefits of olive oil, it is crucial to view it as part of a holistic dietary pattern, not merely as an isolated component. This approach, which integrates olive oil with other nutrient-dense foods and healthy lifestyle habits, is key to achieving optimal metabolic health outcomes [350].

Extensive meta-analyses have demonstrated significant inverse associations between high adherence to the Mediterranean diet and the risk of developing T2DM, as well as improvements in glycemic control among individuals with T2DM following a Mediterranean dietary pattern, compared to a low-fat diet [181]. The Mediterranean diet is not only one of the most widely researched dietary models, but also a highly palatable and sustainable approach [351]. This raises an important question regarding which specific components of the Mediterranean diet may be beneficial when adopted in countries with different dietary traditions, particularly in the absence of suitable local food substitutes. Olive oil, a primary source of dietary fat in the Mediterranean diet, contains unique bioactive components not commonly found in other plant oils. Its high content of monounsaturated fatty acids (MUFA) and various phenolic compounds, such as hydroxytyrosol, tyrosol, and secoiridoids, make olive oil a rich source of antioxidant, anti-inflammatory, insulin-sensitizing, cardioprotective, anti-atherogenic, neuroprotective, and immunomodulatory properties. These bioactive components confer numerous health benefits in the prevention, development, and progression of chronic and age-related diseases, including T2DM [352].

A recent meta-analysis of randomized controlled trials involving patients with T2DM, which included 24 studies with 1460 participants comparing high-MUFA to high-carbohydrate diets and 4 studies with 44 participants comparing high-MUFA to high-polyunsaturated fatty acid (PUFA) diets, showed that consuming diets rich in MUFA can improve metabolic risk factors in T2DM patients [276]. Notably, there were significant reductions in fasting plasma glucose levels when comparing high-MUFA to both high-carbohydrate and high-PUFA diets.

Further, another meta-analysis comprising four cohort studies with 15,784 T2DM cases and 29 trials found that each 10 g daily increase in olive oil consumption was associated with a 9% reduction in the risk of T2DM. However, a nonlinear relationship was observed, where the risk of T2DM decreased by 13% with increasing olive oil intake up to approximately 15–20 g/day, with no additional benefit observed with higher intake levels [181]. Additionally, olive oil supplementation in T2DM patients led to a more pronounced reduction in HbA1c and fasting plasma glucose levels compared to control groups. Given these promising findings, the question arises as to which specific components of olive oil are responsible for these beneficial effects and the underlying molecular mechanisms involved.

The studies reviewed consistently indicate that olive oil components play a pivotal role in enhancing insulin sensitivity by modulating key insulin signaling pathways, regulating lipid metabolism, and reducing oxidative stress and inflammation. Oleic acid, a primary component of olive oil, was shown to improve insulin signaling through the activation of the PI3K-Akt pathway, which is crucial for GLUT4 translocation and glucose uptake. Additionally, oleic acid has been demonstrated to enhance fatty acid oxidation by upregulating the expression of CPT-1A, which facilitates the transport of fatty acids into mitochondria for β-oxidation.

These findings align with earlier studies indicating that oleic acid promotes mitochondrial β-oxidation by upregulating genes involved in fatty acid metabolism, such as peroxisome proliferator-activated receptor gamma coactivator 1-alpha (PGC-1α) and CPT-1, through the activation of the SIRT1-PGC1α transcriptional complex [353,354].

Furthermore, oleic acid has been shown to counteract the negative effects of saturated fatty acids, such as palmitate, on insulin signaling pathways, reinforcing its protective role in metabolic regulation.

The phenolic compounds in olive oil, such as oleuropein and oleacein, have also been implicated in modulating various molecular pathways related to glucose and lipid metabolism, as well as insulin sensitivity. Similar to oleic acid, these phenolic compounds were found to activate the PI3K-Akt pathway. Oleacein, in particular, has been shown to reduce lipogenesis in hepatocytes by downregulating the expression of SREBP-1 and FAS.

Preserving beta cell function is crucial in preventing the progression of T2DM, and olive oil components appear to exert significant protective effects in this regard. The reviewed studies revealed that oleic acid and phenolic compounds, including hydroxytyrosol and tyrosol, protect beta cells from apoptosis induced by saturated fatty acids and endoplasmic reticulum (ER) stress. These compounds reduce the expression of pro-apoptotic markers, such as CHOP and cleaved caspase-3, and downregulate stress-related proteins involved in the unfolded protein response (UPR), such as Bip, ATF6, *p*-eIF2α, and XBP1s, thus helping to preserve beta cell viability and insulin secretion. Moreover, the modulation of oxidative stress and enhancement of autophagic processes in beta cells by oleic acid and other olive oil polyphenols suggest a multifaceted approach to maintaining beta cell function. However, it is important to note that some studies have reported potential lipotoxic effects associated with prolonged oleic acid exposure, highlighting the need for further investigation into the dose-dependent effects of oleic acid on beta cells.

Chronic low-grade inflammation and oxidative stress are key contributors to insulin resistance and beta cell dysfunction in T2DM. Several studies have highlighted the ability of oleic acid to reduce oxidative stress and inflammation, such as by reversing the palmitate-induced activation of the ROS/JUN pathway. The anti-inflammatory and antioxidant properties of olive oil phenolic compounds, including hydroxytyrosol, tyrosol, and secoiridoid derivatives, are well-documented in the reviewed literature. These compounds were shown to inhibit key inflammatory pathways, including the NF-κB pathway, and reduce the expression of pro-inflammatory cytokines (e.g., TNFα, IL-6, IL-1β) and enzymes (e.g., MCP-1, COX-2, iNOS, NOX-1). Additionally, the antioxidant effects of these compounds, evidenced by the reduction in ROS production, highlight their potential to mitigate oxidative stress-related beta cell dysfunction.

The findings from the reviewed studies underscore the potential of olive oil, particularly its bioactive components like oleic acid and polyphenols, to modulate gene expression and molecular mechanisms associated with the pathogenesis and management of T2DM. The substantial role of olive oil in enhancing insulin sensitivity, preserving beta cell function, and mitigating inflammation and oxidative stress underscores its therapeutic potential in addressing the multifactorial nature of T2DM. However, it is crucial to consider the potential variability in the effects of different olive oil components, the influence of dietary context, and the need for further research to fully elucidate the molecular mechanisms underlying its effects and to determine the long-term impacts and optimal use of olive oil in the prevention and management of T2DM.

Although there are many health benefits linked to olive oil, especially in the prevention and treatment of type 2 diabetes, the relationship is not linear. Moderate use of premium olive oil yields the most benefits; greater intakes result in declining returns. This emphasizes how crucial both quantity and quality are when it comes to dietary advice for people who want to lower their risk of type 2 diabetes [181].

More research is required to improve these suggestions and consider the differences in how each person reacts to olive oil, as well as the influence of dietary context and lifestyle factors. At moderate levels of olive oil consumption, there is clear evidence of beneficial effects on insulin sensitivity, glucose metabolism, and cardiovascular health, which can reduce the risk of developing T2DM. However, research suggests that the benefits plateau or even decline at higher consumption levels. This phenomenon could be due to a saturation effect, that is, once an optimal amount of healthy fats is consumed, further increases in intake may provide no additional benefit or may even lead to negative effects, such as excess calorie intake or weight gain [355].

As a potential mechanism, excessive fat intake, even if from healthy sources like olive oil, may contribute to an energy imbalance and lead to obesity, which is a significant risk factor for T2DM. Moreover, an imbalance in the omega-6/omega-3 fatty acid ratio at very high levels of olive oil consumption may lead to increased inflammation and insulin resistance [356].

## 5. Concluding Remarks

While olive oil, particularly extra virgin olive oil (EVOO), has been widely recognized for its health benefits, including its positive effects on type 2 diabetes (T2DM) risk and insulin sensitivity, several limitations in current research must be addressed. Although numerous observational studies and randomized controlled trials (RCTs) have provided valuable insights, notable gaps in knowledge still warrant further investigation.

The absence of long-term human clinical trials is one of the most critical shortcomings in the available data. Most research on how olive oil affects type 2 diabetes and insulin sensitivity is brief and ignores the long-term, cumulative effects of olive oil use on metabolic health. Longitudinal studies are crucial to determine the sustainability of the advantages of olive oil, particularly with chronic conditions like type 2 diabetes, and to provide precise guidelines for the ideal intake levels over time. Long-term trials are resource-intensive and challenging to implement, mainly because they require large sample sizes, controlled conditions, and consistent adherence to olive oil-based diets over extended periods. Additionally, human trials must account for genetic variations, lifestyle factors, and baseline health conditions, which may affect individual responses to olive oil consumption. Understanding the precise mechanisms by which phenolic compounds exert their effects is still an area of active research. There is a need for studies that specifically focus on how these compounds are metabolized in the human body and how individual differences in metabolism may influence the therapeutic potential of olive oil. Furthermore, there is a need for more research into the optimal concentration and forms of olive oil (e.g., unfiltered vs. filtered EVOO) to maximize bioavailability.

Most micronutrients in olive oil are present in very low concentrations, typically around 1 mg per liter, making it unlikely to reach significant levels in the human body. However, bioavailability plays a crucial role in how effectively the body can absorb and utilize these compounds. Some studies suggest that the bioavailability of polyphenols from olive oil can be enhanced when consumed with other fats or in combination with food, which may increase the effective concentration of these compounds in the body [357].

In conclusion, olive oil plays a significant role in modulating the expression of genes and molecular mechanisms involved in the pathogenesis of T2DM, attributed to the synergistic effects of its phenolic compounds and high oleic acid content. These components enhance insulin sensitivity, protect beta cells from apoptosis, and reduce inflammation and oxidative stress, highlighting their therapeutic potential. These findings, supported by clinical evidence, advocate for the inclusion of olive oil in the diet as part of a comprehensive strategy for preventing and managing T2DM.

While the molecular mechanisms of olive oil have primarily been investigated through in vitro experiments and animal models, there is a lack of robust clinical studies. To fully elucidate the role of olive oil in T2DM, further human intervention studies and clinical trials, incorporating genomic, proteomic, and metabolomic data, are necessary. This integrated approach would facilitate a deeper understanding of the biological significance of olive oil’s bioactive components at the cellular, tissue, and organ levels, thus enhancing our comprehension of its health benefits. Moreover, the application of molecular data to predict and evaluate gene responses to olive oil bioactive compounds in personalized nutrition and medicine strategies holds considerable promise and warrants further exploration.

Translating preclinical animal findings into practical dietary recommendations for humans is a complex task, as the effects observed in animals may not fully replicate in humans. More human-based studies that closely mimic real-world dietary patterns and take into account the diversity of human populations are needed. Such studies should consider the doses of olive oil typically consumed in human diets and the interactions between olive oil and other foods consumed in the Mediterranean diet. Therefore, while olive oil holds significant promise as a dietary intervention for T2DM prevention and insulin sensitivity, there remain several critical gaps in current research. By addressing limitations such as the lack of long-term human studies, the variability in bioavailability of its phenolic compounds, and the challenges of translating animal data to humans, we can improve our understanding of how olive oil works to improve metabolic health. As research progresses, it will be crucial to refine dietary recommendations and explore the optimal ways to incorporate olive oil into a health-promoting diet for diverse populations.

### Future Perspective

To address these limitations, future research should focus on conducting long-term human clinical trials to assess the sustainability and effectiveness of olive oil on metabolic health, particularly in relation to T2DM. Additionally, studies should explore the bioavailability of olive oil’s bioactive compounds, considering factors such as genetic differences and gut microbiota composition. Finally, more human-based trials that incorporate a realistic dietary context, reflecting everyday olive oil consumption patterns, are necessary to ensure that findings from animal models can be accurately translated into practical health recommendations for humans.

## Figures and Tables

**Figure 1 nutrients-17-00570-f001:**
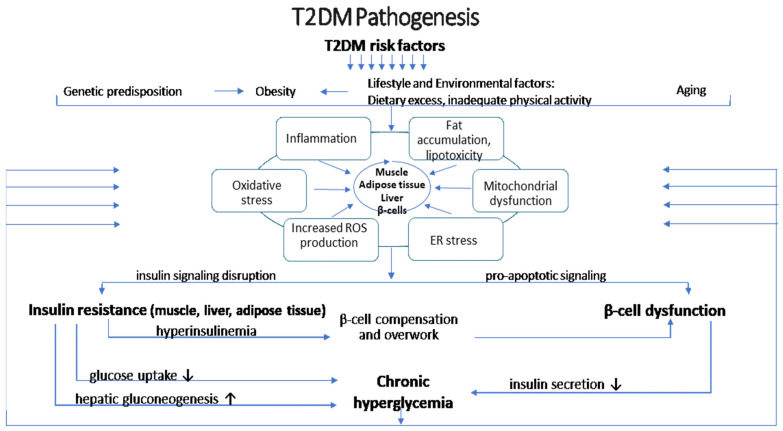
T2DM Pathogenesis: The multifaceted metabolic disorders leading to T2DM.

**Figure 2 nutrients-17-00570-f002:**
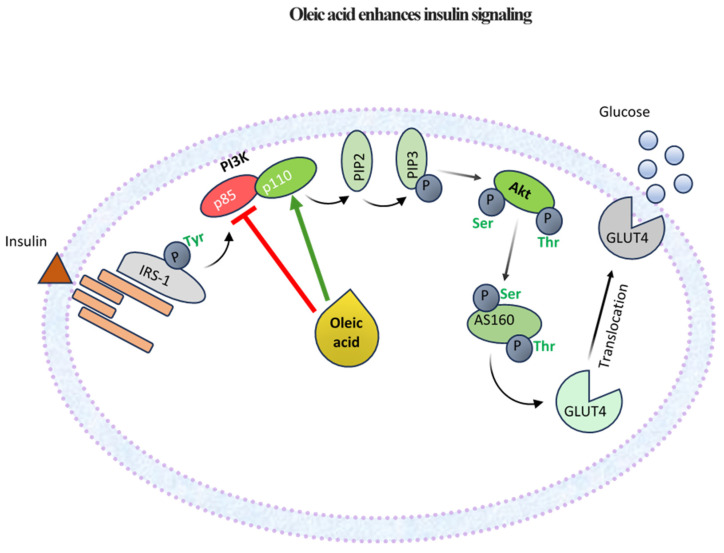
Beneficial effects of oleate on insulin signaling by affecting the expression of PI3K signaling pathway proteins. Oleate, a monounsaturated fatty acid, has a beneficial effect on insulin signaling by altering the PI3K signaling pathway. In a dose-dependent manner, it downregulates p85α, a regulatory subunit of PI3K, and the p85α/p110β ratio while upregulating IRS-1 (insulin receptor substrate 1) and p110β, a catalytic subunit of PI3K. Maximizing PI3K pathway activation promotes glucose metabolism and lessens insulin resistance, which increases insulin sensitivity.

**Figure 3 nutrients-17-00570-f003:**
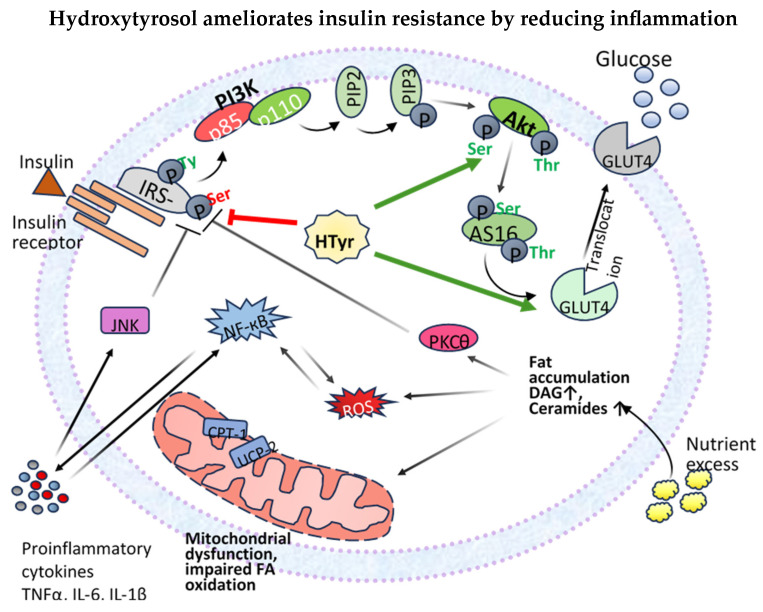
Beneficial effects of hydroxytyrosol (HTyr) on insulin signaling through the enhancement of insulin signaling pathways. The polyphenol hydroxytyrosol (HTyr) in olive oil positively affects insulin signaling by strengthening critical insulin pathways. By stimulating the PI3K/Akt signaling pathway and IRS-1 (insulin receptor substrate 1), it increases insulin sensitivity and facilitates the uptake and metabolism of glucose. Furthermore, HTyr supports improved glucose regulation and metabolic health by lowering oxidative stress and inflammation, which helps minimize insulin resistance.

**Figure 4 nutrients-17-00570-f004:**
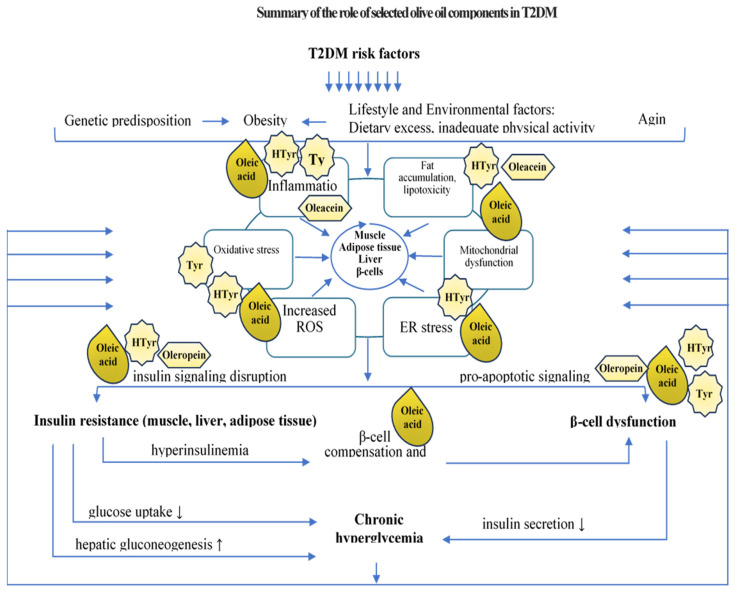
Summary of the role of selected olive oil components in T2DM progression. Specific components of olive oil, notably oleate, and hydroxytyrosol, inhibit the progression of type 2 diabetes mellitus (T2DM) by addressing essential risk factors like β-cell dysfunction, insulin resistance, and persistent hyperglycemia. These compounds decrease inflammation and combat oxidative stress, which are key factors in developing T2DM, and enhance insulin sensitivity by activating insulin signaling pathways. Components of olive oil protect pancreatic β-cells and improve glucose metabolism, slowing the evolution of T2DM and promoting better metabolic health.

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
