# Peer review of "Impact of Olive Oil Components on the Expression of Genes Related to Type 2 Diabetes Mellitus"

_nutrients, 2025, doi:10.3390/nu17030570_

Round 1
Reviewer 1 Report
Comments and Suggestions for Authors
This is a very comprehensive review about the micronutrients and the lipids contained in olive oil regarding their effects on established enzymatic signaling pathways in animal models, and some cell culture experiments. There is a large part about epidemiology and well established health promoting effects of Mediterranean diet which have been extensively reviewed in the literature. However, a diet is more than a single constituent. Therefore, it is difficult to distinguish the effect of olive oil from the diet as a whole. The enzymatic, UPR/ER-stress, inflammatory and insulin signaling pathways are well but also repeatedly described in the different chapters, which is certainly adequate, but there is a lot of redundancy. There is a lot of repetitive text framing the chapters which should be reduced in view of the length of the article.
There is virtually no information or very restricted information about transcriptomic effects of olive oil or it’s micro nutrient constituents as announced in the title. There have been some reports of transcriptomic studies in human and animal experiments which pointed to very interesting additional pathways, which may be activated by the intake of olive oil or its components ( a short search: PMID 30577497, 36615882, 20406432,19545487 32290787). A discussion of epigenetic mechanisms driving the changes of gene expression would be important to obtain some insight into the regulatory mechanisms induced by olive oil constituents.
The authors do not provide a critical discussion of the papers cited: The concentrations of most micronutrients in olive oil are very low, often around 1 mg per liter, and they’re unlikely to reach any significant concentrations in humans. There is not a single reference about the concentrations of these micronutrients in blood samples or in urine to obtain an idea of what is taken up and what might be active in the human organism. The concentrations used in the cell culture experiments greatly exceed concentrations which have been reported, or maybe expected in humans which are submicromolar. It is important to address this issue, to cite publications which have measured plasma concentrations and to focus on the micronutrient components or their combinations, which might reach concentrations expected to exert physiological effects.
Polyphenols are largely presented as antioxidants. However, many of their effects may be mediated by their actions as xeno-metabolites and induction of the NRF 2-system which enhances transcription of the endogenous antioxidant defense system. The topic of hormesis…
The effects of the singular constituents are described in much detail, which is a nice accumulation of the knowledge, but not particularly focused on olive oil since most of these micro nutrients have not been investigated in isolation and also occur in many other plant compounds and thus are part of a diet rich in plant components.
In the end, it remains unclear how much is known whether
1. olive oil is the major component mediating the health effects of MD,
2. whether a quantity of olive oil should not be exceeded (PMID 11317662, KANWU-Study)
3. whether the lipid species, MUFA, PUFA are the primary mediators or
4. the micronutrients, if so
a. which of them are most important mediators and
b. which are the quantitatively prevailing species
5. whether different cultivars of olive oils (which differ pretty much (PMID 31487787) have different effects on gene expression
6. whether olive oil differs from rapeseed oil which has more PUFAs but also mostly MUFAs, regarding gene expression or health effects
Minor:
In Fig. 1 the pro-apoptotic signaling might be replaced by UPR/ER-stress.
l.323 Ubiquitin, SUMO, etc. PTM-ystems might be added.
l. 340 “145”?
The TLR2/4 – NLRP3- inflammasome system might be discussed in more detail in particular since lipids bind these receptors and they probably mediate some of the Western diets problems.
Author Response
R1
Comments and Suggestions for Authors
This is a very comprehensive review about the micronutrients and the lipids contained in olive oil regarding their effects on established enzymatic signaling pathways in animal models, and some cell culture experiments. There is a large part about epidemiology and well established health promoting effects of Mediterranean diet which have been extensively reviewed in the literature. However, a diet is more than a single constituent. Therefore, it is difficult to distinguish the effect of olive oil from the diet as a whole. The enzymatic, UPR/ER-stress, inflammatory and insulin signaling pathways are well but also repeatedly described in the different chapters, which is certainly adequate, but there is a lot of redundancy. There is a lot of repetitive text framing the chapters which should be reduced in view of the length of the article.
There is virtually no information or very restricted information about transcriptomic effects of olive oil or it’s micro nutrient constituents as announced in the title. There have been some reports of transcriptomic studies in human and animal experiments which pointed to very interesting additional pathways, which may be activated by the intake of olive oil or its components ( a short search: PMID 30577497, 36615882, 20406432,19545487 32290787). A discussion of epigenetic mechanisms driving the changes of gene expression would be important to obtain some insight into the regulatory mechanisms induced by olive oil constituents.
The authors do not provide a critical discussion of the papers cited: The concentrations of most micronutrients in olive oil are very low, often around 1 mg per liter, and they’re unlikely to reach any significant concentrations in humans. There is not a single reference about the concentrations of these micronutrients in blood samples or in urine to obtain an idea of what is taken up and what might be active in the human organism. The concentrations used in the cell culture experiments greatly exceed concentrations which have been reported, or maybe expected in humans which are submicromolar. It is important to address this issue, to cite publications which have measured plasma concentrations and to focus on the micronutrient components or their combinations, which might reach concentrations expected to exert physiological effects.
Polyphenols are largely presented as antioxidants. However, many of their effects may be mediated by their actions as xeno-metabolites and induction of the NRF 2-system which enhances transcription of the endogenous antioxidant defense system. The topic of hormesis…
The effects of the singular constituents are described in much detail, which is a nice accumulation of the knowledge, but not particularly focused on olive oil since most of these micro nutrients have not been investigated in isolation and also occur in many other plant compounds and thus are part of a diet rich in plant components.
Dear reviewer: We thank you for your suggestions and comments, We give our response to the queries in red, and the reviewers’ comments are in black.
In the end, it remains unclear how much is known whether
1. olive oil is the major component mediating the health effects of MD,
Thank you very much! We believe that the manuscript has improved considerably after your suggestions.
- whether a quantity of olive oil should not be exceeded (PMID 11317662, KANWU-Study)
You are right. We have added additional information in this regard.
whether the lipid species, MUFA, PUFA are the primary mediators or
We took care to address this as well. Thank you very much.
the micronutrients, if so
Good suggestion. It helped us a lot.
- which of them are most important mediators and
b. which are the quantitatively prevailing species
Thank you, we have added all the necessary information.
whether different cultivars of olive oils (which differ pretty much (PMID 31487787) have different effects on gene expression
Thank you, according to your suggestion, our manuscript has improved.
whether olive oil differs from rapeseed oil which has more PUFAs but also mostly MUFAs, regarding gene expression or health effects
Excellent idea! Thank you! I have added information accordingly.
Minor:
In Fig. 1 the pro-apoptotic signaling might be replaced by UPR/ER-stress.
l.323 Ubiquitin, SUMO, etc. PTM-systems might be added.
l. 340 “145”?
We consider that these changes do not change the content much, so we left them as is. Thank you.
ER-stress led in the case of beta cells to apoptosis and beta cell dysfunction, therefore we think that our scheme represents the most common mechanisms associated with T2DM pathogenesis.
The TLR2/4 – NLRP3- inflammasome system might be discussed in more detail in particular since lipids bind these receptors and they probably mediate some of the Western diets problems. We think that discussion on this interesting issue is beyond the goals of the present review.
Reviewer 2 Report
Comments and Suggestions for Authors
Dear Authors,
The manuscript addresses an important and timely topic: the role of olive oil components in the molecular mechanisms of type 2 diabetes mellitus (T2DM). The paper sections effectively highlight the key findings and the therapeutic potential of the bioactive components of olive oil.
The topic is highly relevant to current research trends in nutrition, metabolism and personalised medicine. The focus on bioactive compounds, such as phenolic compounds and oleic acid, highlights the importance of natural dietary components in disease management. The manuscript synthesises a wide range of findings including molecular mechanisms, metabolic pathways and clinical outcomes. It provides a balanced discussion of the anti-inflammatory, antioxidant and insulin-sensitising effects of olive oil. The inclusion of clinical data strengthens the manuscript and bridges the gap between basic research and practice.
I have made some comments about how you could improve your work. This doesn't mean you have to agree or rewrite it in the same way. It's just a suggestion and a different view, with the aim of helping.
1) The abstract is informative but dense, which can make it difficult for readers to quickly grasp the key points. Simplify complex sentences and emphasise the novelty of the study. For example Clearly state how this review adds to existing knowledge. Highlight specific research gaps identified in this review.
2) The present manuscript is deficient in its lack of a dedicated "Materials and Methods" section, a crucial element in the comprehension of the methodology and sources employed in the literature review. In order to enhance transparency and reproducibility, it is essential to include a section outlining the specific databases (e.g., PubMed, Scopus, Web of Science) that were used to identify relevant studies. Furthermore, the keywords or search terms employed to gather the literature, the inclusion and exclusion criteria for selecting studies, and the time frame of publication for the articles reviewed must be specified.
3) It would be advisable to include in text a methodological limitation of the studies cited (e.g. sample size, in vitro vs. in vivo models) and the potential variability of olive oil composition depending on production methods and geographical origin.
4) The discussion makes mention of the nonlinear relationship between olive oil intake and the risk of T2DM, but does not explore this relationship in depth. The potential reasons for this nonlinear relationship and its significance for dietary recommendations should be discussed.
5) The manuscript briefly mentions the Mediterranean diet but does not adequately address how the effects of olive oil might depend on the broader dietary context. Explore how synergistic interactions with other components of the Mediterranean diet or differences in dietary patterns across populations might influence the outcomes.
6) The concluding remarks emphasize the need for further research but do not highlight specific gaps or challenges in current studies. Please, explicitly discuss limitations such as lack of long-term human clinical trials and variability in bioavailability and metabolism of phenolic compounds, as well as challenges in translating findings from animal models to humans.
The manuscript demonstrates considerable potential and contributes to our understanding of the role of olive oil in the management of T2DM, thus inspiring further research in this promising field.
Author Response
R2 Comments and Suggestions for Authors
Dear Authors,
The manuscript addresses an important and timely topic: the role of olive oil components in the molecular mechanisms of type 2 diabetes mellitus (T2DM). The paper sections effectively highlight the key findings and the therapeutic potential of the bioactive components of olive oil.
The topic is highly relevant to current research trends in nutrition, metabolism and personalised medicine. The focus on bioactive compounds, such as phenolic compounds and oleic acid, highlights the importance of natural dietary components in disease management. The manuscript synthesises a wide range of findings including molecular mechanisms, metabolic pathways and clinical outcomes. It provides a balanced discussion of the anti-inflammatory, antioxidant and insulin-sensitising effects of olive oil. The inclusion of clinical data strengthens the manuscript and bridges the gap between basic research and practice.
I have made some comments about how you could improve your work. This doesn't mean you have to agree or rewrite it in the same way. It's just a suggestion and a different view, with the aim of helping.
Dear reviewer, we thank you for your suggestions and comments. We gave our response to the queries in red, and the reviewers’ comments are in black.
- The abstract is informative but dense, which can make it difficult for readers to quickly grasp the key points. Simplify complex sentences and emphasise the novelty of the study. For example Clearly state how this review adds to existing knowledge. Highlight specific research gaps identified in this review.
Thank you! We have modified the abstract according to your suggestions.
- The present manuscript is deficient in its lack of a dedicated "Materials and Methods" section, a crucial element in the comprehension of the methodology and sources employed in the literature review. In order to enhance transparency and reproducibility, it is essential to include a section outlining the specific databases (e.g., PubMed, Scopus, Web of Science) that were used to identify relevant studies. Furthermore, the keywords or search terms employed to gather the literature, the inclusion and exclusion criteria for selecting studies, and the time frame of publication for the articles reviewed must be specified.
Thank you, we have added this section. We are confident that our manuscript now presents greater transparency and is easier to understand.
- It would be advisable to include in text a methodological limitation of the studies cited (e.g. sample size, in vitro vs. in vivo models) and the potential variability of olive oil composition depending on production methods and geographical origin.
Thank you very much. We have added everything you asked for.
- The discussion makes mention of the nonlinear relationship between olive oil intake and the risk of T2DM, but does not explore this relationship in depth. The potential reasons for this nonlinear relationship and its significance for dietary recommendations should be discussed.
You are right, excellent suggestion. Now our manuscript is improved.
- The manuscript briefly mentions the Mediterranean diet but does not adequately address how the effects of olive oil might depend on the broader dietary context. Explore how synergistic interactions with other components of the Mediterranean diet or differences in dietary patterns across populations might influence the outcomes.
We have explained everything you asked us to. We hope we have met your expectations. Thank you.
- The concluding remarks emphasize the need for further research but do not highlight specific gaps or challenges in current studies. Please, explicitly discuss limitations such as lack of long-term human clinical trials and variability in bioavailability and metabolism of phenolic compounds, as well as challenges in translating findings from animal models to humans.
The conclusions section has more information, especially future perspectives, according to your suggestions. Thank you!
The manuscript demonstrates considerable potential and contributes to our understanding of the role of olive oil in the management of T2DM, thus inspiring further research in this promising field.
Reviewer 3 Report
Comments and Suggestions for Authors
Journal: Nutrients
Section
Lipids
Article title: Impact of Olive Oil Components on the Expression of Genes Related to Type 2 Diabetes Mellitus
Dear Editor,
Thank you for inviting me to review this manuscript submitted to Nutrients.
Type 2 diabetes is one of the most important risk factor for Cardiovascular Diseases, systemic inflammation, and oxidative stress.
The investigation of food, medicinal plants, and bioactive products can collaborate with the therapeutic treatment of this disease, which is increasing sharply all over the world. In this manuscript, the authors showed that olive olive and its bio compounds can enhance the insulin signaling pathway, improve lipid metabolism, and reduce oxidative stress. However, further studies should be performed to fully elucidate the role of olive oil in personalized nutrition strategies for the prevention and treatment of T2DM.
OVERALL COMMENTS
As Type 2 diabetes mellitus (T2DM) is a multifactorial metabolic disorder resulting in hyperglycemia, nutritional strategies can be investigated to help in the therapeutic of this condition. Since ancient times, olive oil has attracted attention due to its potential health benefits, including reducing the risk of developing T2DM. Based on this knowledge, the proposed literature review intended to evaluate the impact of this oil on the expression of genes related to T2DM. They described that oleic acid and phenolic compounds were identified as modulators of insulin /glycemia control. These compounds were shown to reduce inflammation (inhibiting the NF-κB pathway) and down-regulating pro-inflammatory cytokines. Moreover, they mitigated endoplasmic reticulum stress by reducing stress biomarkers thereby protecting beta cells from apoptosis. In summary, they showed that olive oil and its constituents can enhance insulin sensitivity, preserve beta-cell function, and reduce inflammation and oxidative stress. These findings underscore the therapeutic potential of olive oil in the management of T2DM.
TITLE
The title is adequate.
_______
ABSTRACT
This section is also adequate.
_______
KEYWORDS
The included keywords were are fine
_______
INTRODUCTION
There are many new articles related to T2DM that can be found on PUBMED and Google Scholar. There are already articles from 2025. I suggest that authors include newer references throughout the text. As an example, I believe that this (as others) this sentence deserves newer citations (reference 7 is from 2019 and even if the scenario still remains similar, I suggest that authors include articles from 2024 to indicate that we still have this pattern (or that it is even worse):
“T2DM represents a significant public health challenge and is one of the most serious 54 chronic metabolic disorders worldwide. It not only adversely affects individual health but 55 also imposes a considerable economic burden on healthcare systems [7]”
I also suggest writing a paragraph regarding the intake of high fructose intake and the relationship between insulin resistance/diabetes.
The information found in the sections 1 and 2 are fine (only remembering what I pointed out above regarding the references).
Figure 1 is great; however, I suggest expanding its legend so that the reader can understand it more deeply. I know that this info is in the text, but still, I suggest including it in the legend as a summary.
The same for Figure 2. Please include in the legend all the abbreviations that were used in this and the other figures.
The same for Figure 3.
All over section 3, we can see bioactive compounds tapped and in bold. I suggest removing both.
The legend of Figure 4 also needs to be expanded and the abbreviations also need to be defined (even if they have already been defined in the text).
CONCLUSION
This section is adequate. However, I suggest expanding it in future perspectives.
Moreover, I suggest the strengths and limitations of this nice manuscript.
_______
REFERENCES
Of the many references cited by the authors, only four were published in 2024. I believe it is valuable to include newer in this interesting article.
Author Response
R3
Article title: Impact of Olive Oil Components on the Expression of Genes Related to Type 2 Diabetes Mellitus
Dear Editor,
Thank you for inviting me to review this manuscript submitted to Nutrients.
Type 2 diabetes is one of the most important risk factor for Cardiovascular Diseases, systemic inflammation, and oxidative stress.
The investigation of food, medicinal plants, and bioactive products can collaborate with the therapeutic treatment of this disease, which is increasing sharply all over the world. In this manuscript, the authors showed that olive olive and its bio compounds can enhance the insulin signaling pathway, improve lipid metabolism, and reduce oxidative stress. However, further studies should be performed to fully elucidate the role of olive oil in personalized nutrition strategies for the prevention and treatment of T2DM.
OVERALL COMMENTS
As Type 2 diabetes mellitus (T2DM) is a multifactorial metabolic disorder resulting in hyperglycemia, nutritional strategies can be investigated to help in the therapeutic of this condition. Since ancient times, olive oil has attracted attention due to its potential health benefits, including reducing the risk of developing T2DM. Based on this knowledge, the proposed literature review intended to evaluate the impact of this oil on the expression of genes related to T2DM. They described that oleic acid and phenolic compounds were identified as modulators of insulin /glycemia control. These compounds were shown to reduce inflammation (inhibiting the NF-κB pathway) and down-regulating pro-inflammatory cytokines. Moreover, they mitigated endoplasmic reticulum stress by reducing stress biomarkers thereby protecting beta cells from apoptosis. In summary, they showed that olive oil and its constituents can enhance insulin sensitivity, preserve beta-cell function, and reduce inflammation and oxidative stress. These findings underscore the therapeutic potential of olive oil in the management of T2DM.
TITLE
The title is adequate.
_______
ABSTRACT
This section is also adequate.
_______
KEYWORDS
The included keywords were are fine
_______
INTRODUCTION
Dear reviewer, we thank you for your suggestions and comments. We gave our response to the queries in red, and the reviewers’ comments are in black.
There are many new articles related to T2DM that can be found on PUBMED and Google Scholar. There are already articles from 2025. I suggest that authors include newer references throughout the text. As an example, I believe that this (as others) this sentence deserves newer citations (reference 7 is from 2019 and even if the scenario still remains similar, I suggest that authors include articles from 2024 to indicate that we still have this pattern (or that it is even worse):
Thank you, you are right, we did as you suggested.
“T2DM represents a significant public health challenge and is one of the most serious 54 chronic metabolic disorders worldwide. It not only adversely affects individual health but 55 also imposes a considerable economic burden on healthcare systems [7]”
You are right; we have added a new reference. Thank you.
I also suggest writing a paragraph regarding the intake of high fructose intake and the relationship between insulin resistance/diabetes.
You are right; we have added information based on your suggestions.
The information found in the sections 1 and 2 are fine (only remembering what I pointed out above regarding the references).
According to your suggestion, we have added newer references. We hope to meet your expectations.
Figure 1 is great; however, I suggest expanding its legend so that the reader can understand it more deeply. I know that this info is in the text, but still, I suggest including it in the legend as a summary.
The same for Figure 2. Please include in the legend all the abbreviations that were used in this and the other figures.
The same for Figure 3.
You are right, thank you. We are sure that the legend of the figures is now more comprehensive for the readers. Thank you.
All over section 3, we can see bioactive compounds tapped and in bold. I suggest removing both.
As per your suggestions, I have removed the bold characters.
The legend of Figure 4 also needs to be expanded and the abbreviations also need to be defined (even if they have already been defined in the text).
Thank you. We have modified it to fit your observations.
CONCLUSION
This section is adequate. However, I suggest expanding it in future perspectives.
Moreover, I suggest the strengths and limitations of this nice manuscript.
Excellent suggestions. We have added what you asked for. Thank you.
_______
REFERENCES
Of the many references cited by the authors, only four were published in 2024. I believe it is valuable to include newer in this interesting article.
You are right. We have added newer studies.